

# Intermittency of gravity wave potential energies and absolute momentum fluxes derived from infrared limb sounding satellite observations

Manfred Ern[1], Peter Preusse[1], and Martin Riese[1]

[1]Institut für Energie- und Klimaforschung – Stratosphäre (IEK–7), Forschungszentrum Jülich GmbH, 52425 Jülich, Germany

**Correspondence:** M. Ern (m.ern@fz-juelich.de)

**Abstract.** Atmospheric gravity waves contribute significantly to the driving of the global atmospheric circulation. Because of their small spatial scales, their effect on the circulation is usually parameterized in general circulation models. These parameterizations, however, are strongly simplified. One important effect that is often neglected is the fact that gravity wave sources, and thus the global distribution of gravity waves, are both very intermittent. Therefore, time series of global observations of

5 gravity waves are needed to study the distribution, seasonal variation, and strength of this effect.

For gravity wave absolute momentum fluxes and potential energies observed by the limb sounding satellite instruments High Resolution Dynamics Limb Sounder (HIRDLS) and Sounding of the Atmosphere using Broadband Emission Radiometry (SABER), we investigate the global distribution of gravity wave intermittency by deriving probability density functions (PDFs) in different regions, as well as global distributions of Gini coefficients. In the stratosphere, we find that intermittency is strongest

10 in mountain wave regions, followed by the polar night jets, and regions of deep convection in the summertime subtropics. Intermittency is weakest in the tropics. A better comparability of intermittency in different years and regions is achieved by normalizing single observations by their monthly median distributions. Our results are qualitatively in agreement with previous findings from satellite observations, and quantitatively in good agreement with previous findings from superpressure balloons and high resolution models. Generally, momentum fluxes exhibit stronger intermittency than potential energies, and lognormal

15 distributions are often a reasonable approximation of the PDFs. In the tropics, we find that, for monthly averages, intermittency increases with altitude, which might be a consequence of variations in the atmospheric background, and thus varying gravity wave propagation conditions. Different from this, in regions of stronger intermittency, particularly in mountain wave regions, we find that intermittency decreases with altitude, which is likely related to the dissipation of large-amplitude gravity waves during their upward propagation.

## 1 Introduction

Atmospheric waves are important drivers of the atmospheric circulation (e.g., Andrews et al., 1987, and references therein). Particularly, the global observation and modeling of gravity waves is very challenging because of their small spatial scales (e.g., Fritts and Alexander, 2003, and references therein). In the middle atmosphere, typical horizontal wavelengths of gravity





waves are in the range of a few tens to a few thousand kilometers. Their vertical wavelengths are in the range from below $1\,\mathrm{km}$ to several ten kilometers (e.g., Preusse et al., 2008; Alexander et al., 2010, and references therein).

Many gravity wave sources are located in the troposphere and lower stratosphere. Some of the most relevant gravity wave sources are atmospheric flow over topography (e.g., McFarlane, 1987; Lott and Miller, 1997; Eckermann and Preusse, 1999; Kruse et al., 2022), deep convection (e.g., Fovell et al., 1992; Pfister et al., 1993; Piani et al., 2000; Song and Chun, 2005; Stephan et al., 2019a, b; Ern et al., 2022), and processes related to strong wind jets and fronts (e.g., Charron and Manzini, 2002; Zhang, 2004; Zülicke and Peters, 2006; Plougonven and Zhang, 2014; Kim et al., 2016; Wei et al., 2016; Geldenhuys et al., 2021). According to their sources, these waves are also called mountain waves (or orographic gravity waves), convective gravity waves, and jet- or front-generated gravity waves, respectively.

Gravity waves propagate away from their sources, both vertically and horizontally, redistribute energy and momentum in the atmosphere, and thereby couple different atmospheric layers and regions. When gravity waves dissipate, they exert forcing ("gravity wave drag") on the atmospheric background flow (e.g., McLandress, 1998; Fritts and Alexander, 2003).

The vertical flux of horizontal pseudomomentum of a gravity wave (for simplification, in the following just denoted "momentum flux") is given by:

$$(F_{px}, F_{py}) = \overline{\varrho}\ \left(1 - \frac{f^2}{\widehat{\omega}^2}\right) (\overline{u'w'},\ \overline{v'w'}) \tag{1}$$

(e.g., Fritts and Alexander, 2003), with $F_{px}$ and $F_{py}$ the zonal and the meridional momentum flux, respectively, $\overline{\varrho}$ the atmospheric background density, $f$ the Coriolis frequency, $\widehat{\omega}$ the intrinsic frequency of the gravity wave, and $u'$, $v'$, and $w'$ the wind perturbations of the background atmosphere in zonal, meridional, and vertical direction, respectively, that are caused by the gravity wave. Averaging over one or multiple full wave cycles is indicated by the overbars. The absolute momentum flux $F_{ph}$ of a gravity wave is given by:

$$F_{ph} = \sqrt{F_{px}^2 + F_{py}^2}, \tag{2}$$

and the gravity wave drag $(X, Y)$ that a gravity wave exerts on the background flow is given by:

$$(X, Y) = -\frac{1}{\overline{\varrho}} \frac{\partial (F_{px}, F_{py})}{\partial z} \tag{3}$$

with $X$ and $Y$ the gravity wave drag in zonal and meridional direction, respectively, and $z$ the vertical coordinate. Another important parameter is the gravity wave potential energy $E_{pot}$:

$$E_{pot} = \frac{1}{2} \left(\frac{g}{N}\right)^2 \overline{\left(\frac{T'}{\overline{T}}\right)^2}, \tag{4}$$

with $T'$ the temperature fluctuations due to the gravity wave, $g$ the gravity acceleration, $N$ the buoyancy frequency, and $\overline{T}$ the atmospheric background temperature. Using the gravity wave temperature amplitude $\widehat{T}$, this can be rewritten as follows:

$$E_{pot} = \frac{1}{4} \left(\frac{g}{N}\right)^2 \left(\frac{\widehat{T}}{\overline{T}}\right)^2. \tag{5}$$



See also Ern et al. (2018).

Gravity waves contribute significantly to the driving of the meridional circulation in the stratosphere (e.g., Alexander and
Rosenlof, 2003) and in the mesosphere (e.g., Holton, 1983). They are the main drivers of the wind reversal at the top of the
mesospheric wind jets in both the summer hemisphere and the winter hemisphere (e.g., Lindzen, 1981; Holton, 1982). Gravity
wave drag is also an important contribution to the zonal momentum budget in the tropics, and, together with global-scale waves,
they are driving the quasi-biennial oscillation (QBO) (e.g., Lindzen and Holton, 1968; Ern and Preusse, 2009a, b; Alexander
and Ortland, 2010; Ern et al., 2014) and the semiannual oscillation (SAO) (e.g., Delisi and Dunkerton, 1988; Antonita et al.,
2007; Ern et al., 2015; Smith et al., 2022) of the zonal wind in the tropics. Further, gravity waves contribute to the variations
of the polar night jets around sudden stratospheric warmings (SSWs) (e.g., Hitchcock and Shepherd, 2013; Albers and Birner,
2014; Ern et al., 2016) and contribute to the forcing of global-scale waves in the mesosphere (e.g., Holton, 1984; Smith, 2003;
Ern et al., 2013; Matthias and Ern, 2018; Sato et al., 2018).

Another important effect is that temperature fluctuations of gravity waves contribute to the formation of ice clouds and thus
dehydration in the upper troposphere and the tropopause region (e.g., Schoeberl et al., 2015; Dinh et al., 2016), as well as to
the formation of polar stratospheric clouds (e.g., Carslaw et al., 1999; Eckermann et al., 2009), and thus to ozone depletion in
the polar regions (e.g., Orr et al., 2020, and references therein).

General circulation models and chemistry climate models (GCMs/CCMs) usually resolve only parts of the whole spectrum
of gravity waves. Therefore, the effect of gravity waves on the global circulation is simulated by gravity wave parameteri-
zations (e.g., Fritts and Alexander, 2003; Kim et al., 2003; Geller et al., 2013, and references therein). These gravity wave
parameterizations are usually very simplified. For example, they assume that gravity waves propagate only vertically, while in
the real atmosphere gravity waves can propagate not only vertically, but also horizontally (e.g., Sato et al., 2009; Preusse et al.,
2009b; Kalisch et al., 2014; Hindley et al., 2015; Thurairajah et al., 2017).

Several parameterizations exist that are dedicated to specific gravity wave source processes. Some examples are McFarlane
(1987) or Lott and Miller (1997) for mountain waves, Charron and Manzini (2002) or de la Cámara and Lott (2015) for gravity
waves exited by jets and fronts, and Beres et al. (2004), Song and Chun (2005), or Bushell et al. (2015) for convectively
generated gravity waves. Many gravity wave parameterizations, however, comprise the contribution of non-orographic gravity
waves into just one parameterization that assumes a globally constant (e.g., Warner and McIntyre, 2001; Orr et al., 2010),
piecewise constant (e.g., Molod et al., 2015), or otherwise very simplified source distribution, even though it is evident that a
more realistic middle atmosphere can be simulated by more realistic gravity wave source distributions (e.g., de la Cámara et
al., 2014; Yigit et al., 2021).

In particular, simplified parameterizations do not account for the intermittency of gravity wave sources and the resulting
intermittent global distributions of gravity waves. In the real atmosphere, gravity wave amplitudes, wavelengths, and momen-
tum fluxes can vary strongly, both spatially and temporally. Particularly, large-amplitude gravity waves will saturate earlier and
exert their drag at different locations than small-amplitude gravity waves (e.g., Fritts, 1984). As a consequence, if uniform and
constant launch amplitudes are assumed, the resulting global distribution of gravity wave drag will not be fully realistic.





To overcome this limitation, several gravity wave parameterizations simulate the intermittency of gravity wave sources by introducing stochastic variations of the gravity wave sources (e.g., Eckermann, 2011; Lott et al., 2012; de la Cámara et al., 2014; de la Cámara and Lott, 2015; Serva et al., 2018). It has been shown, for example, by de la Cámara et al. (2014) that this can lead to more realistic simulations of the QBO, and also the gravity wave forcing at the top of the mesospheric wind jets does not unrealistically peak around a single altitude, but over a range of altitudes, including also somewhat lower altitudes.

Generally, parameterizations need guidance by observations to become more realistic. Consequently, observations of gravity waves and their intermittency are needed to improve stochastic gravity wave parameterizations. In addition, these kind of observations are needed for comparison with gravity wave parameterizations that are dedicated to specific source processes like orography and convection, as well as for comparison with gravity waves that are explicitly resolved by high-resolution models. Some examples of gravity wave intermittency observations from ground based stations are Zink and Vincent (2001), Cao and Liu (2016), Minamihara et al. (2020), or Conte et al. (2022). Gravity wave intermittency was studied also using superpressure balloon observations in the Southern Hemisphere at polar latitudes (Hertzog et al., 2008, 2012; Plougonven et al., 2013; Jewtoukoff et al., 2015), as well as in the tropics (Jewtoukoff et al., 2013; Corcos et al., 2021). Further, gravity wave intermittency was derived from satellite observations, for example, from global navigation satellite system radio occultations (GNSS-RO) (Baumgaertner and McDonald, 2007), from High Resolution Dynamics Limb Sounder (HIRDLS) observations (Hertzog et al., 2012; Wright et al., 2013), and from nadir soundings of the Atmospheric Infrared Sounder (AIRS) instrument (Alexander and Grimsdell, 2013; Wright et al., 2017).

In our study, we determine gravity wave intermittency for monthly global distributions of gravity wave potential energies and absolute momentum fluxes derived from observations of the limb sounding satellite instruments HIRDLS and Sounding of the Atmosphere using Broadband Emission Radiometry (SABER). Compared to previous estimates of gravity wave intermittency from limb sounders, our data covers a larger range of gravity wave vertical wavelengths. In addition, SABER observations cover a larger altitude range, including the whole mesosphere. This allows to follow the evolution of gravity wave intermittency from the mid stratosphere (close to the gravity wave sources) to the upper mesosphere where gravity waves strongly dissipate and drive the reversal of the mesospheric wind jets.

The instruments HIRDLS and SABER are briefly introduced in Sect. 2, and in Sect. 3 we describe how gravity wave potential energies and absolute momentum fluxes are derived. In Sect. 4, gravity wave intermittency is discussed based on probability density functions (PDFs). In Sect. 5, we introduce the Gini coefficient for investigating global distributions of intermittency in the stratosphere with better spatial resolution. Using distributions of Gini coefficients, the evolution of intermittency in the vertical direction is investigated in Sect. 6. Finally, Sect. 7 gives a summary and discussion.

## 2 The satellite instruments HIRDLS and SABER

The satellite instruments HIRDLS and SABER observe Earth's atmosphere in limb-viewing geometry. HIRDLS was launched onboard the Earth Observing System (EOS) Aura satellite and provided observations from 22 January 2005 until 17 March 2008 in the latitude range from about 63°S to 80°N. SABER was launched onboard the Thermosphere Ionosphere Mesosphere





Energetics and Dynamics (TIMED) satellite and started atmospheric observations on 25 January 2002. SABER measurements are still ongoing at the time of writing. The TIMED satellite performs yaw maneuvers every about 60 days. As a consequence, SABER changes between a northward-viewing and a southward-viewing measurement geometry every about 60 days for about 60 days. The latitude coverages are about 50°S to 82°N and about 82°S to 50°N, respectively. Therefore, the latitude coverage of monthly averages is either 50°S to 82°N or 82°S to 50°N for those months not containing a yaw maneuver, or, in the case

of months containing a yaw, 82°S to 82°N, but with coverage at high latitudes only during part of the month. Initially, yaws were performed during "odd" months (i.e., January, March, May, July, September, and November), but the times of the yaw maneuvers have gradually shifted during the SABER mission.

While both instruments observe several atmospheric trace species, our work focuses on HIRDLS and SABER temperature observations. Both instruments are infrared radiometers, and atmospheric temperatures are derived from infrared emissions of

$CO_2$ at around 15 $\mu$m. Because both instruments are limb sounders, they provide temperature altitude profiles with good vertical resolution. The vertical resolution is about 1 km for HIRDLS, and about 2 km for SABER. The altitude range of HIRDLS temperatures is from about the tropopause to near the mesopause. The SABER instrument was designed for observations at even higher altitudes, and temperatures cover the altitude range from about the tropopause to well above 100 km.

An instrument description of the HIRDLS instrument is given in Gille et al. (2003), and the HIRDLS temperature retrieval

is described in Gille et al. (2008, 2011). The SABER instrument is described in more detail in Mlynczak (1997) and Russell et al. (1999). More information on the SABER temperature retrieval can be found in Remsberg et al. (2004, 2008).

## 3 Gravity wave analysis based on satellite limb soundings

### 3.1 Determination of gravity wave temperature fluctuations

Observed temperature altitude profiles are a superposition of the large-scale atmospheric background and of the temperature

fluctuations due to gravity waves. We isolate the temperature fluctuations due to gravity waves by following the approach described in Ern et al. (2018). First, from each altitude profile a zonal average altitude profile is subtracted. The resulting altitude profiles of residual temperatures still contain the contributions of both global-scale waves and of gravity waves.

The contribution of global-scale waves is determined by a dedicated spectral analysis (Ern et al., 2011). Two-dimensional spectra in longitude and time are calculated in overlapping time windows of 31 days length for a set of fixed altitudes and

latitudes. The contribution of global-scale waves is determined for each observation in every altitude profile from these spectra for the respective location and time. This approach removes global-scale waves with periods as short as about 1.3 days and covers two-day waves, tropical Kelvin waves, and inertial instabilities that are difficult to remove by other methods (e.g., Ern et al., 2008, 2009; Ern et al., 2013; Rapp et al., 2018; Strube et al., 2020). Additional high-pass filtering was applied separately to each altitude profile in order to limit the range of vertical wavelengths still contained in each altitude profile to shorter than

about 25 km. In this way, remnants of global-scale waves are further reduced, and the range of gravity wave vertical wavelengths is limited to the range that is suitable for the momentum flux analysis described in Sect. 3.2.





In an additional step, atmospheric tides are removed from the temperature residuals. For satellites in slowly precessing low Earth orbits, solar tides appear as wave patterns that are stationary if ascending (satellite flying southward) and descending (satellite flying southward) orbit parts are considered separately in time intervals that are much shorter than the period of one

full satellite precession cycle. The reason is that during these short time intervals ascending and descending parts of the satellite orbit are measured at different and about constant local solar time. This fact is utilized to remove tides up to apparent zonal wavenumber 4 from observed altitude profiles (Ern et al., 2013).

After performing the above mentioned steps, the resulting altitude profiles of residual temperatures can be attributed mainly to gravity waves. An approximation for the sensitivity of limb sounding satellite instruments for gravity waves was derived by

Preusse et al. (2002) as a function of gravity wave horizontal and vertical wavelength. Approximate sensitivity functions that apply to the HIRDLS and SABER data sets used here are given in Ern et al. (2018).

### 3.2 Gravity wave potential energies and absolute momentum fluxes

For deriving gravity wave absolute momentum fluxes from temperature observations, Eq. (1) has to be rewritten in terms of gravity wave temperature amplitudes using the linear gravity wave polarization relations (Ern et al., 2004; Ern et al., 2017)

$$(F_{px}, F_{py}) = \frac{1}{2}\overline{\varrho}\left(\frac{g}{N}\right)^2 \frac{(k, l)}{m}\left(\frac{\widehat{T}}{\overline{\overline{T}}}\right)^2 \tag{6}$$

which involves the 3D gravity-wave wave-vector with $k$, $l$, and $m$ the zonal, meridional, and vertical wavenumber, respectively. For gravity wave absolute momentum fluxes, we obtain:

$$F_{ph} = \frac{1}{2}\overline{\varrho}\left(\frac{g}{N}\right)^2 \frac{\lambda_z}{\lambda_h}\left(\frac{\widehat{T}}{\overline{\overline{T}}}\right)^2 \tag{7}$$

with $k_h = \left(k^2 + l^2\right)^{0.5} = 2\pi/\lambda_h$ the horizontal wavenumber of the gravity wave, $\lambda_h$ its horizontal wavelength, and $\lambda_z$ its

vertical wavelength.

For each observed vertical profile of residual temperatures, we carry out a combination of maximum entropy method and harmonic analysis (MEM/HA) after Preusse et al. (2002) to derive altitude profiles of gravity wave amplitudes, vertical wavelengths, and phases for the strongest wave component at each altitude, based on running 10 km vertical windows. These gravity wave amplitudes are used to calculate gravity wave potential energies after Eq. (5). This can be performed for each altitude

profile.

The estimation of gravity wave absolute momentum fluxes is more difficult because the gravity wave horizontal wavelength $\lambda_h$ has to be estimated. For gravity wave momentum fluxes, we follow the approach of Ern et al. (2004, 2011) and focus on pairs of altitude profiles along the satellite measurement track that are horizontally apart by no more than about 300 km. For these pairs of altitude profiles, we determine the vertical phase difference of the strongest wave at each altitude. From these

phase differences, the apparent horizontal wavelength of a gravity wave parallel to the satellite measurement track can be estimated. For an illustration see, for example, Preusse et al. (2009a) and Ern et al. (2018).





We assume that the same wave is seen in both altitude profiles if the vertical wavelengths in the two profiles differ by no more than 40%. All other pairs of altitude profiles are neglected, i.e. those with non-matching vertical wavelength, and those that have too large distances between the two profiles. The remaining pairs of altitude profiles are likely still representative for the whole distribution of gravity waves, because distributions of gravity wave squared amplitudes of the remaining pairs are approximately equal to the distributions calculated from single altitude profiles, and also approximately equal to the distributions calculated from the unused pairs of altitude profiles (see also Ern et al., 2018).

Absolute momentum fluxes are calculated by assuming that the horizontal wavelength parallel to the measurement track can be used as a proxy for the true horizontal wavelength of a gravity wave. Because the horizontal wavelength parallel to the measurement track will always overestimate the true horizontal wavelength of a gravity wave, this will introduce large biases and likely result in an underestimation of absolute momentum fluxes. Other error sources are aliasing effects caused by an undersampling of observed gravity waves, and effects caused by the instrument sensitivity functions. Both these effects could cause an even stronger underestimation of absolute momentum fluxes. The full observational filter of limb sounding satellite instruments is discussed in more detail by, for example, Trinh et al. (2015), or Trinh et al. (2016). Overall errors of the so-derived momentum fluxes are at least a factor of two (see also Ern et al., 2004; Ern et al., 2017, 2018).

As an example, Fig. 1 shows global distributions of gravity wave potential energies for the HIRDLS instrument at an altitude of 30 km for each calendar month. The time range used for averaging is from March 2005 until February 2008. The global distributions were gridded using overlapping bins of 15° longitude × 5° latitude. Figure 2 shows the same, but for the SABER instrument using an averaging period from January 2002 until October 2020 and, because SABER has a coarser sampling, larger longitude latitude bins of 15° longitude × 10° latitude are used. Different from the global distributions shown in Ern et al. (2018), values are multi-year averages of medians, and not averages. Further, for each single month entering the multi-year averages, grid points are not used if fewer than 40 data points are contained in the respective grid box.

Similar as Figs. 1 and 2, Figs. 3 and 4 show global distributions of median absolute gravity wave momentum fluxes at 30 km altitude for each calendar month for HIRDLS (Fig. 3) and SABER (Fig. 4). For HIRDLS, we again use overlapping bins of 15° longitude × 5° latitude. For SABER, however, due to the reduced number of available data points, we use a coarser resolution of 30° longitude × 20° latitude, i.e. worse than for SABER gravity wave potential energies.

The global distributions shown in Figs. 1–4 are the result of seasonally varying gravity wave sources and seasonally varying gravity wave propagation conditions given by the background winds and background temperature profile. In case of potential energies also seasonal variations of the background density are important (for a discussion see also Strelnikova et al., 2021). In the subtropics of the respective summer hemisphere, characteristic enhancements of gravity wave activity are found that are likely caused by gravity waves excited by deep convection over the continents, as well as over the Maritime Continent. At mid and high latitudes of the respective winter hemisphere, we find strong gravity wave activity in the polar night jets and their vicinity. Partly, these gravity waves are excited by jet-related source processes. Partly, mountain waves excited by flow over mountain ranges form hot spots, for example, over South America, the Antarctic Peninsula, Scandinavia, or Greenland. Overall, the global distributions of medians display relative variations that are very similar to the global distributions of averages shown in Ern et al. (2018).





**Figure 1.** Global distributions of HIRDLS gravity wave (GW) potential energies at 30 km altitude for each calendar month. Values shown are multi-year averages of monthly median values determined in overlapping 15° longitude × 5° latitude grid boxes. Period used for averaging is March 2005 until February 2008.





**Figure 2.** Global distributions of SABER gravity wave (GW) potential energies at 30 km altitude for each calendar month. Values shown are multi-year averages of monthly median values determined in overlapping 15° longitude × 10° latitude grid boxes. Period used for averaging is January 2002 to October 2020.





**Figure 3.** Same as Fig. 1, but for median HIRDLS gravity wave (GW) absolute momentum fluxes.





**Figure 4.** Same as Fig. 2, but for median SABER gravity wave (GW) absolute momentum fluxes determined in 30° longitude × 20° latitude grid boxes.

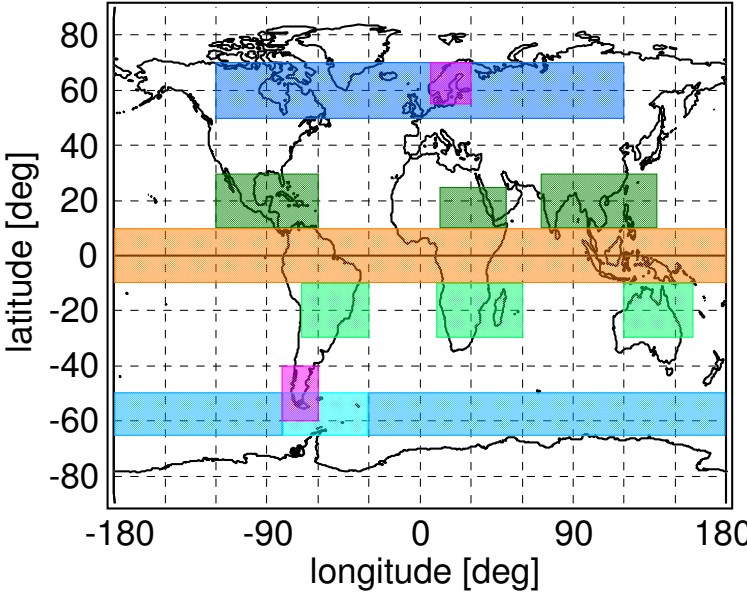

**Figure 5.** Illustration of the different regions selected for creating PDFs. The longitude/latitude ranges of the different regions are summarized in Table 1.

## 4 Gravity wave intermittency investigated by probability density functions (PDFs)

The global distributions in Figs. 1–4 show regular patterns that are similar in different years. In spite of these robust patterns, gravity wave activity is very intermittent, both spatially and temporally. Whenever intermittency of gravity wave distributions is determined, this requires collecting data temporally and/or spatially in a certain time-interval/region. In our case, we collect data over one month in several pre-defined longitude/latitude intervals for a given altitude. Of course, these choices will have effect on the level of intermittency that is obtained in our analysis, as both temporal variations of gravity wave sources and of gravity wave propagation conditions within these intervals will contribute. The longitude/latitude regions selected in our work for determining PDFs are illustrated in Fig. 5, and the corresponding longitude/latitude ranges are summarized in Table 1.

### 4.1 A first example: PDFs at southern hemisphere mid/high latitudes in October

One method to investigate the intermittency of the gravity wave distribution are probability density functions (PDFs). As an example, Fig. 6a displays for the latitude range 50–65°S and all longitudes the absolute momentum flux PDF for the HIRDLS instrument at 30 km altitude, separately for the months October 2005 (blue), October 2006 (red), and October 2007 (blue-green). For all PDFs shown in Fig. 6, the mean, the 90[th] percentile, the 99[th] percentile, as well as the fractions of the mean momentum fluxes at values beyond the respective percentiles are given in Table 2.

The region 50–65°S is dominated by the Southern Ocean (flat terrain) except for South America (see also Fig. 5), which means that most gravity waves detected in this region are likely not mountain waves. Like in Hertzog et al. (2012), we find that



**Table 1.** Latitude/longitude ranges of the different regions illustrated in Fig. 5.

| region | latitude range | longitude range |
|---|---|---|
| tropics | 10°S–10°N | 180°W–180°E |
| NH subtropics (1) | 10°N–30°N | 120°W–60°W |
| NH subtropics (2) | 10°N–25°N | 10°E–50°E |
| NH subtropics (3) | 10°N–30°N | 70°E–140°E |
| SH subtropics (1) | 10°S–30°S | 30°W–70°W |
| SH subtropics (2) | 10°S–30°S | 10°E–60°E |
| SH subtropics (3) | 10°S–30°S | 120°E–160°E |
| southern mid/high latitudes (Fig. 6) | 50°S–65°S | 180°W–180°E |
| Southern Ocean (no orography) | 50°S–65°S | 180°W–80°W and 30°W–180°E |
| South America | 40°S–60°S | 60°W–80°W |
| northern mid/high latitudes | 50°N–70°N | 120°W–120°E |
| Scandinavia | 55°N–70°N | 5°E–30°E |

**Table 2.** Means, 90[th] percentiles, 99[th] percentiles, as well as fractions of gravity wave absolute momentum fluxes at values beyond the respective percentiles for the PDFs of momentum fluxes and normalized momentum fluxes shown in Fig. 6 for the month of October in the latitude band 50–65°S.

| respective PDF | mean momentum flux | 90[th] percentile | >90[th] percentile | 99[th] percentile | >99[th] percentile |
|---|---|---|---|---|---|
| Fig. 6a, HIRDLS, October 2005 | 1.62 mPa | 3.38 mPa | 57.6% | 18.0 mPa | 21.5% |
| Fig. 6a, HIRDLS, October 2006 | 2.28 mPa | 5.15 mPa | 56.3% | 23.7 mPa | 17.6% |
| Fig. 6a, HIRDLS, October 2007 | 1.46 mPa | 2.99 mPa | 60.2% | 16.4 mPa | 24.2% |
| Fig. 6b, HIRDLS, October 2005 | 2.32 | 4.99 | 52.3% | 23.2 | 17.2% |
| Fig. 6b, HIRDLS, October 2006 | 2.57 | 5.60 | 54.9% | 26.7 | 18.2% |
| Fig. 6b, HIRDLS, October 2007 | 2.25 | 4.91 | 50.5% | 19.3 | 16.6% |
| Fig. 6c, HIRDLS, October combined | 2.38 | 5.19 | 52.1% | 22.6 | 17.4% |
| Fig. 6e, SABER, z=30 km, October combined | 2.36 | 5.31 | 47.4% | 19.3 | 13.2% |
| Fig. 6f, SABER, z=80 km, October combined | 1.52 | 3.51 | 37.8% | 9.01 | 8.12% |





**Figure 6.** Probability density functions (PDFs) of **(a)** HIRDLS gravity wave absolute momentum fluxes (MF) over the Southern Ocean (latitudes 65–50°S, all longitudes) at 30 km altitude, for the month of October in the years 2005 (blue) 2006 (red), and 2007 (blue-green). **(b)** Same as (a), but momentum fluxes were normalized by the monthly median global distribution. **(c)** Same as (b), but all years were combined into one PDF. **(d)** Same as (c), but on a logarithmic scale. **(e)** Same as (c), but for the SABER instrument and combining the October values for the years 2002–2020. **(f)** Same as (e), but on a logarithmic scale. **(g)** Same as (e), but for an altitude of 80 km. **(h)** Same as (g), but on a logarithmic scale. Red curves that are overplotted in all panels are the corresponding lognormal distributions.





the PDFs follow a lognormal distribution for each of the months. The respective lognormal distributions, which are character-
ized by the mean and the standard deviation of the logarithmic momentum flux values, are indicated by the smooth curves in
the color of the respective year. The fact that the distributions are roughly lognormal means that the PDFs have a long tail at
high momentum fluxes, and the largest 10% (1%) of values contribute as much as about 60% (20%) to the average momentum
flux in the region. This finding is similar as for the HIRDLS data in Hertzog et al. (2012) (their Fig. 2).

It is however notable that the average, median, and 90$^{th}$ percentile momentum fluxes in different years are different, while
the shape of the PDFs is very similar. This means that if one wants to combine data from different years, the PDFs should
be created from single values that are normalized by, for example, the global distribution of median values. Normalization by
median values makes particularly sense if PDFs are expected to follow a lognormal distribution, as the median characterizes the
center of a lognormal distribution. Normalization of distributions may be particularly important in the tropics and subtropics
where the QBO modulates the gravity wave distribution, in addition to seasonal variations (e.g., Ern et al., 2011; Ern et al.,
2014; Chen et al., 2019). Normalization of the PDFs makes also sense if only the shapes of the PDFs of different data sets (that
potentially have different sensitivities for gravity waves and thus different levels of gravity wave momentum fluxes or potential
energies) shall be compared. This applies for observations, as well as for model data.

Figure 6b shows the same as Fig. 6a, but the single momentum flux observations were normalized by the October global
distribution of medians of the respective year. As can be seen from Fig. 6b, the PDFs of the different years are almost on top of
each other, further demonstrating that the statistical properties of the momentum flux distributions in different years are very
similar. Also the relative contributions of momentum fluxes at values beyond particularly the 99$^{th}$ percentiles are more similar
for the normalized momentum fluxes (see Table 2). Further, the relative contributions of momentum fluxes at values beyond
the respective percentiles are usually somewhat lower than those for the unnormalized momentum fluxes, indicating that local
variations of momentum fluxes contribute to the intermittency of the PDFs for the unnormalized values. This again shows the
advantage of using normalized values for PDFs.

It should also be noted that in Hertzog et al. (2008) the ratio of the median (i.e., the 50$^{th}$ percentile) and the 90$^{th}$ percentile
was introduced as a measure of intermittency. As the median of "normalized" PDFs is very close to unity, the reciprocal of
the 90$^{th}$ percentile of a "normalized" PDF can be directly taken as a measure of intermittency: the higher the 90$^{th}$ percentile
of a "normalized" PDF, the stronger the intermittency of the distribution. This is a very practical application of using normal-
ized values for creating PDFs. In particular, the 90$^{th}$ percentiles of "normalized" PDFs of different parameters, and thus the
intermittency of the different parameters (for example, gravity wave potential energies and absolute momentum fluxes), can be
directly compared.

As the distributions in different years are very similar, we combine the normalized gravity wave momentum flux values of
the three PDFs into a single PDF. The result is shown in Fig. 6c. Again, the red curve represents the corresponding lognormal
distribution. As can be seen from Table 2, the mean, the 90$^{th}$ and 99$^{th}$ percentiles, and the momentum flux relative contributions
beyond the respective percentiles are close to the values of the single years.

Most previous studies displayed PDFs only on a linear scale, which made it difficult to investigate the shape of the PDF at
low momentum flux values. To overcome this shortcoming, we also display in Fig. 6d the PDF of Fig. 6c on a logarithmic scale.





As can be seen from Fig. 6d, the PDF follows a lognormal distribution also for a large range of 1–2 magnitudes of normalized momentum fluxes at values below zero, i.e. at values lower than the location of the distribution maximum. Only at the very
lowest values the PDF starts to exceed the lognormal distribution.

Figures 6e and 6f show the same as Figures 6c and 6d, but for the SABER instrument and combining the October data of the years 2002 until 2020. Obviously, the SABER PDFs in Figs. 6c and 6d are very similar to those of the HIRDLS instrument. Only at the very highest momentum fluxes the SABER PDFs decrease more strongly than the HIRDLS PDFs. Likely reason is the coarser along-track sampling of SABER, which leads to stronger aliasing and stronger overestimation of the horizontal
wavelength (i.e., underestimation of absolute momentum fluxes) of short horizontal wavelength gravity waves that potentially carry large momentum fluxes. This is also reflected in the reduced numbers of the 90th and 99th percentiles and the momentum flux relative contributions beyond the respective percentiles given in Table 2.

Figures 6g and 6h show the same as Figures 6e and 6f, but for an altitude of 80 km. Compared to 30 km altitude, the distribution at 80 km is more strongly skewed toward low values. This is expected for two reasons: Firstly, the along-track sampling
distance of those pairs of SABER altitude profiles that are used for calculating momentum fluxes increases with altitude, leading to stronger undersampling (aliasing) of gravity wave horizontal wavelengths and thus to low-biases of gravity wave momentum fluxes (cf. Ern et al., 2011). Secondly, large-amplitude gravity waves that potentially carry stronger momentum fluxes will reach saturation at lower altitudes, dissipate, and are thus removed from the PDF. This also leads to much reduced numbers of the 90th and 99th percentiles, as well as to reduced relative contributions of the momentum fluxes beyond the
respective percentiles (see Table 2).

### 4.2 PDFs of gravity wave absolute momentum fluxes and potential energies for specific regions

In the following, we will investigate the characteristics of the PDFs in different regions. The locations of the different regions are illustrated in Fig. 5. We focus on the HIRDLS instrument because of the better HIRDLS along-track sampling. Similar as in Figs. 6c and 6d, we again combine the momentum fluxes of all years available for HIRDLS to improve statistics.

### 290   4.2.1 Gravity wave momentum flux PDFs in the tropics and in the respective summer hemisphere

Figure 7 shows PDFs of HIRDLS gravity wave absolute momentum fluxes at 30 km altitude in different regions in the tropics and the respective summer hemisphere. The first row in Fig. 7 shows PDFs for the tropics in the latitude band 10°S–10°N and all HIRDLS observations from January 2005 until February 2008. Shown are the PDF for unnormalized momentum fluxes on a linear scale (Fig. 7a), also on a linear scale the PDF of momentum fluxes normalized by the global distribution of median
momentum fluxes for each respective month that enters the PDF (Fig. 7b), and in Fig. 7c the same as Fig. 7b, but on a logarithmic scale. The mean, the 90th percentile, the 99th percentile, as well as the fractions of the mean momentum fluxes at values beyond the respective percentiles for the different PDFs in Fig. 7 are given in Table 3.

Generally, the shapes of the unnormalized PDFs are quite similar to the PDFs composed of normalized momentum fluxes (and potential energies, see Sect. 4.2.3 below). The distributions of unnormalized momentum fluxes are only generally some-





**Figure 7.** PDFs of HIRDLS gravity wave absolute momentum fluxes (MF) at 30 km altitude in the tropics for the period from January 2005 until February 2008 ((**a**)–(**c**)), for the three hotspots of gravity wave activity in the subtropics of the northern hemisphere during boreal summer (JJA) ((**d**)–(**f**)), for the three hotspots of gravity wave activity in the subtropics of the southern hemisphere during austral summer (DJF) ((**g**)–(**i**)), for the northern hemisphere mid/high latitude region 50°N–70°N, 120°W–120°E during boreal summer (JJA) ((**j**)–(**l**)), and for the Southern Ocean region 50°S–65°N without the longitudes 80°W–30°W during austral summer (DJF) ((**m**)–(**o**)). For an illustration of the locations of the different regions see Fig. 5. The left column shows gravity wave absolute momentum flux in mPa, while the middle and right colums show relative momentum flux.





**Table 3.** Means, 90[th] percentiles, 99[th] percentiles, as well as fractions of gravity wave absolute momentum fluxes at values beyond the respective percentiles for the PDFs of momentum fluxes (upper part of the table) and normalized momentum fluxes (lower part of the table) shown in Figs. 7 and 8.

| region | mean momentum flux | 90[th] percentile | >90[th] percentile | 99[th] percentile | >99[th] percentile |
|---|---|---|---|---|---|
| tropics (all months) | 0.61 mPa | 1.19 mPa | 34.7% | 2.72 mPa | 6.87% |
| NH subtropics (JJA) | 1.41 mPa | 3.27 mPa | 37.6% | 8.22 mPa | 7.76% |
| SH subtropics (DJF) | 1.07 mPa | 2.48 mPa | 38.6% | 6.38 mPa | 8.44% |
| 50–70°N, 120°W–120°E (JJA) | 0.42 mPa | 0.97 mPa | 38.5% | 2.54 mPa | 8.58% |
| Southern Ocean (DJF) | 0.30 mPa | 0.70 mPa | 40.1% | 1.97 mPa | 9.66% |
| Southern Ocean (JJA) | 4.70 mPa | 11.1 mPa | 50.2% | 41.2 mPa | 13.7% |
| South America (JJA) | 10.3 mPa | 23.2 mPa | 67.3% | 133.5 mPa | 21.4% |
| 50–70°N, 120°W–120°E (DJF) | 2.10 mPa | 4.84 mPa | 49.0% | 18.0 mPa | 14.1% |
| Scandinavia (DJF) | 2.87 mPa | 6.86 mPa | 50.6% | 26.0 mPa | 15.0% |
| region | mean normalized momentum flux | 90[th] percentile | >90[th] percentile | 99[th] percentile | >99[th] percentile |
| tropics (all months) | 1.45 | 3.27 | 33.9% | 7.16 | 6.02% |
| NH subtropics (JJA) | 1.55 | 3.51 | 36.7% | 8.41 | 7.05% |
| SH subtropics (DJF) | 1.61 | 3.76 | 37.5% | 9.02 | 7.90% |
| 50–70°N, 120°W–120°E (JJA) | 1.57 | 3.51 | 36.7% | 8.81 | 7.91% |
| Southern Ocean (DJF) | 1.66 | 3.85 | 38.6% | 10.1 | 9.16% |
| Southern Ocean (JJA) | 2.22 | 5.19 | 48.6% | 18.4 | 13.0% |
| South America (JJA) | 3.81 | 8.41 | 62.9% | 48.4 | 19.5% |
| 50–70°N, 120°W–120°E (DJF) | 1.94 | 4.42 | 44.6% | 14.6 | 12.0% |
| Scandinavia (DJF) | 2.04 | 4.52 | 47.2% | 17.0 | 13.6% |

what wider. However, the unnormalized PDFs and values in the different regions are not directly comparable in terms of their absolute values. Therefore, the following discussion will focus on the PDFs based on normalized values.

From the PDFs shown in Figs. 7a–7c, it is evident that in the tropics the tail of the PDFs at large momentum fluxes is far below the red curve that represents a lognormal distribution, and the PDF is skewed toward low momentum flux values. This means that the distribution of gravity wave momentum fluxes in the tropics is much less intermittent than the distribution at

high latitudes during winter (cf. Fig. 6). This is in agreement with previous findings by Wright et al. (2013) and Ern et al. (2014) for HIRDLS momentum fluxes. Also the 90[th] percentiles are in good agreement with previous findings in the tropics (Wright et al., 2013; Ern et al., 2014; Corcos et al., 2021). (Please note that, different from Jewtoukoff et al. (2013), Corcos et al. (2021) did not use superpressure ballon observations above nighttime clouds that lead to high-biased momentum fluxes and possibly to overly long tails at high momentum fluxes in tropical gravity wave momentum flux PDFs.)



Figures 7d–7i show the PDFs for the gravity wave hotspot regions in the summertime subtropics (cf. Figs. 1–4) that were previously discussed by Ern and Preusse (2012). Even in these regions of enhanced gravity wave activity during the summer months, the characteristics of the PDFs are similar as in the tropics: intermittency is only somewhat enhanced compared to the tropics, as can be seen from the 90th and 99th percentiles of the normalized PDFs given in Table 3.

       Also the PDFs shown in Figs. 7j–7o, which represent mid and high latitude regions in the respective summer hemisphere,
are skewed towards low values. The normalized PDFs are very similar to those in the summertime subtropics, and also intermittency is similar as for the distributions in the summertime subtropics, as can be seen from the comparable numbers of the 90th and 99th percentiles for the normalized PDFs given in Table 3.

       In the tropics and summertime subtropics, it is expected that deep convection is one of the main gravity wave source processes (e.g., Beres et al., 2004; Song and Chun, 2005; Kang et al., 2017, 2018). Infrared limb sounding satellite instruments can
only observe gravity waves of horizontal wavelengths longer than 100–200 km (intrinsic periods longer than 1 to 2 hours, cf. Alexander et al. (2010)). These long horizontal wavelengths are attributable rather to mesoscale convective systems than to single convective cells that usually excite gravity waves of shorter horizontal scales and shorter intrinsic periods (e.g., Trinh et al., 2016). Therefore, it should be pointed out that our work focuses only on the part of the gravity wave spectrum seen by infrared limb sounders, and the intermittency of gravity waves can be different in different parts of the gravity wave spectrum.
In particular, superpressure observations indicate that convective gravity waves of intrinsic periods shorter than 1 hour are more intermittent than convective gravity waves of longer intrinsic periods (Corcos et al., 2021).

### 4.2.2    Gravity wave momentum flux PDFs in the respective winter hemisphere

Figure 8 shows PDFs of HIRDLS gravity wave absolute momentum fluxes at 30 km altitude for mid and high latitudes in the respective winter hemisphere. Figures 8a–8c show PDFs for gravity wave momentum fluxes in the austral winter season
June until August over the Southern Ocean, i.e., the latitude band 50–65°S without the longitude range 30–80°W. By omitting the longitude range 30–80°W the gravity wave hotspot over South America is excluded that is likely dominated by mountain waves. In this way, we focus on the intermittency of nonorographic gravity waves over the Southern Ocean. Compared to the situation in October shown in Fig. 6, the PDFs over the Southern Ocean shown in Figs. 8a–8c are very similar, but decrease somewhat more strongly than lognormal at high values. Likely reason is that the very high gravity wave momentum fluxes
of the hotspot over South America are not included in the PDFs shown in Figs. 8a–8c. As can be seen from Tables 2 and 3, the 90th and 99th percentiles for normalized momentum fluxes during June to August are very similar to the percentiles of the corresponding distributions shown in Fig. 6, while the momentum flux fraction at values beyond the 99th percentile is somewhat higher for the corresponding distributions shown in Fig. 6.

       Because the likely reason for the above mentioned differences are the orographic gravity waves over South America, we will
therefore next investigate the PDFs of the gravity wave hotspot over South America. Figures 8d–8f show PDFs for gravity wave momentum fluxes in the austral winter season June until August in the region 40–60°S / 60–80°W, i.e. the region of the gravity wave hotspot over South America that is likely dominated by mountain waves. As was stated before by, for example, Hertzog et al. (2008), Hertzog et al. (2012), and Wright et al. (2013), gravity waves excited by orographic sources are very intermittent.

**Figure 8.** PDFs of HIRDLS gravity wave absolute momentum fluxes (MF) at 30 km altitude over the Southern Ocean during austral winter (JJA) (**(a)**–**(c)**), over the region of the gravity wave hotspot over South America during austral winter (JJA) (**(d)**–**(f)**), over the region of enhanced gravity wave activity in the northern hemisphere polar night jet during boreal winter (DJF) (**(g)**–**(i)**), and over Scandinavia and its near vicinity during boreal winter (DJF) (**(j)**–**(l)**). For an illustration of the locations of the different regions see Fig. 5. The left column shows gravity wave absolute momentum flux in mPa, while the middle and right colums show relative momentum flux.





This is the case because this source mechanism depends on the strongly variable near-surface winds. As can be seen from
Fig. 8e, indeed, the PDF exceeds a lognormal distribution at relative momentum fluxes in the range of about 10–30. Also the
90[th] and 99[th] percentiles of the relative momentum fluxes over South America are the highest values to be found in Table 3.
However, the PDF does not exceed the lognormal distribution as strongly as was found for the momentum fluxes observed by
superpressure balloons, or simulated by high resolution models (e.g., Hertzog et al., 2008, 2012). At relative momentum fluxes
above 30 the PDFs in Figs. 8e and 8f even drop below the lognormal distribution. This is likely an effect of the observational
filter that applies for limb sounding satellite instruments. The very highest momentum fluxes are likely carried by gravity waves
of quite short horizontal wavelength. However, horizontal wavelengths shorter than about 100 km cannot be seen by HIRDLS
and SABER. In addition, short horizontal wavelengths that are still seen by HIRDLS and SABER will suffer from amplitude
low-biases by the sensitivity function of limb sounders (see for example Preusse et al., 2002; Ern et al., 2018, and references
therein), as well as an undersampling and thus overestimation of horizontal wavelengths, resulting in low-biased momentum
fluxes.

In order to find out whether there are hemispheric differences, we will investigate PDFs of the gravity waves in the Northern
Hemisphere at mid and high latitudes during boreal winter. Figures 8g–8i show PDFs for gravity wave momentum fluxes in
the boreal winter season December until February in the region 50–70°N / 120°W–120°E, which corresponds to enhanced
values of gravity wave activity related to the northern hemisphere polar night jet (see also Figs. 1–4). Obviously, the PDFs
of normalized momentum fluxes in Figs. 8h and 8i are very similar to those over the Southern Ocean (Figs. 8b and 8c). The
90[th] and 99[th] percentiles of normalized momentum fluxes, however, are somewhat lower than over the Southern Ocean, which
indicates that the gravity wave distribution in the northern hemisphere polar night jet region is less intermittent on average.

Hotspots of gravity wave activity in the Northern Hemisphere during boreal winter that are linked to orography are usually
less pronounced than the gravity wave hotspot over South America during austral winter (see Figs. 1–4). One of the reasons
might be that, due to the more pronounced large-scale orography, Rossby wave activity in the Northern Hemisphere is usually
stronger. Therefore, it is expected that jet-related gravity source processes act more continuously, and hotspots of mountain
wave activity are therefore swamped with gravity waves from other sources. Further, it has been shown that mountain waves
can be advected over large distances, and in the stratosphere do not necessarily occur over the mountain range where they were
excited (e.g., Krisch et al., 2017). Still, mountain waves are often observed near their sources, for example over Scandinavia
(e.g., Doernbrack and Leutbecher, 2001; Gisinger et al., 2020). Therefore, to include also a region in the Northern Hemisphere
where mountain waves are repeatedly observed, Figs. 8j–8l show PDFs for gravity wave momentum fluxes in the boreal winter
season December until February in the region 55–70°N / 5°E–30°E, which roughly corresponds to Scandinavia and its near
vicinity.

Indeed, for relative momentum fluxes in the range 10–20 the PDFs shown in Figs. 8k and 8l are somewhat closer to a
lognormal distribution than those shown in Figs. 8h and 8i. Further, the PDF in Fig. 8l has a less pronounced tail at low
values of relative momentum fluxes than the PDF in Fig. 8i. However, at high values of relative momentum fluxes the PDFs
in Figs. 8k and 8l do not show an enhancement as strong as seen in the South America region (cf. Figs. 8e and 8f). This
means that the PDFs over Scandinavia are an intermediate state between those PDFs dominated by nonorographic gravity





wave sources (Southern Ocean and northern hemisphere polar night jet regions) and PDFs that are more strongly dominated
by orographic sources (South America region). This might indicate that over Scandinavia often a mixture of mountain waves
and nonorographic waves is observed. Indeed, several case studies show that nonorographic gravity waves are frequently seen
over Scandinavia (e.g., Réchou et al., 2013; Krisch et al., 2020).

### 4.2.3 PDFs of gravity wave potential energies

In addition to gravity wave momentum fluxes, also gravity wave potential energies are of interest for comparison with other
instruments, or with model data. Therefore, Figs. 9 and 10 show PDFs for the same regions as in Figs. 7 and 8, but for
gravity wave potential energies. Similar as Table 3, Table 4 shows the $90^{th}$ and $99^{th}$ percentiles of the PDFs, as well as the
corresponding fractions of potential energies at values beyond the respective percentiles.

Similarly as the PDFs for gravity wave absolute momentum fluxes, the PDFs for gravity wave potential energies roughly
follow lognormal distributions. This has been pointed out before by, for example, Baumgaertner and McDonald (2007). As is
evident from Figs. 7–10, the potential energy PDFs in the different regions are very similar to the corresponding momentum
flux PDFs. This is noteworthy, as one might expect that the shape of the momentum flux PDFs could be skewed by biases
introduced by the satellite sampling and the corresponding biases in horizontal wavelength estimates. Obviously, however,
these biases have no strong effect on the overall shape of the PDFs.

The main difference between potential energy PDFs and momentum flux PDFs is that the potential energy PDFs are gen-
erally narrower than the corresponding momentum flux PDFs. This is also reflected in the lower numbers of $90^{th}$ and $99^{th}$
percentiles of relative potential energy PDFs, compared to PDFs for relative momentum fluxes. The same holds for the frac-
tions of relative momentum fluxes and relative potential energies at values beyond the respective percentiles. The reduced
numbers for potential energy PDFs indicate that potential energy distributions are less intermittent than absolute momentum
flux distributions. This finding is as expected because in the calculation of momentum fluxes also gravity wave horizontal and
vertical wavelengths enter. These additional parameters will add further variability, and thus they contribute to the stronger
intermittency of momentum fluxes compared to potential energies.

### 4.2.4 Effect of measurement noise

The intermittency of gravity wave potential energies is almost exclusively introduced by the intermittency of gravity wave
temperature squared amplitudes (cf. Eq. (5)). This allows us to investigate how reliable might be the long tails of PDFs at low
potential energies (and momentum fluxes). Generally, measurement noise should result in a high-bias of temperature squared
amplitudes (and thus potential energies) of weak gravity wave events.

For an example, we will now roughly estimate to which extent the parts of PDFs at low values may be affected by mea-
surement noise. At $30\,\mathrm{km}$ altitude, the precision $\sigma$ of HIRDLS and SABER temperature observations is about $\sigma=0.4\,\mathrm{K}$ for
HIRDLS and $\sigma=0.3\,\mathrm{K}$ for SABER (e.g., Gille et al., 2011; Ern et al., 2018, and references therein). In our study, we determine
temperature amplitudes $\widehat{T}$ in sliding vertical windows of $10\,\mathrm{km}$ extent. According to the respective vertical field of view, n=10
(n=5) independent values enter an amplitude estimate for HIRDLS (SABER), which reduces the noise-induced uncertainty by

**Figure 9.** Same as Fig. 7, but for gravity wave potential energies $E_{pot}$.





**Figure 10.** Same as Fig. 8, but for gravity wave potential energies $E_{pot}$.





**Table 4.** Means, 90[th] percentiles, 99[th] percentiles, as well as fractions of gravity wave potential energies at values beyond the respective percentiles for the PDFs of potential energies (upper part of the table) and normalized potential energies (lower part of the table) shown in Figs. 9 and 10.

| region | mean potential energies | 90[th] percentile | >90[th] percentile | 99[th] percentile | >99[th] percentile |
|---|---|---|---|---|---|
| tropics (all months) | 2.10 J/kg | 4.32 J/kg | 28.2% | 8.61 J/kg | 5.10% |
| NH subtropics JJA | 3.72 J/kg | 8.04 J/kg | 32.7% | 18.4 J/kg | 6.13% |
| SH subtropics DJF | 3.08 J/kg | 6.53 J/kg | 32.6% | 15.0 J/kg | 6.56% |
| NH 50–70°N 120°W–120°E JJA | 0.77 J/kg | 1.60 J/kg | 32.9% | 3.76 J/kg | 6.73% |
| Southern Ocean DJF | 0.64 J/kg | 1.33 J/kg | 33.7% | 3.27 J/kg | 7.17% |
| Southern Ocean JJA | 8.35 J/kg | 18.8 J/kg | 39.1% | 53.1 J/kg | 9.27% |
| South America JJA | 13.1 J/kg | 27.9 J/kg | 57.8% | 153.1 J/kg | 17.4% |
| NH 50–70°N 120°W–120°E DJF | 5.64 J/kg | 13.0 J/kg | 41.2% | 39.4 J/kg | 9.89% |
| Scandinavia DJF | 7.93 J/kg | 18.4 J/kg | 44.7% | 58.2 J/kg | 11.5% |
| region | mean normalized potential energies | 90[th] percentile | >90[th] percentile | 99[th] percentile | >99[th] percentile |
| tropics (all months) | 1.27 | 2.54 | 28.1% | 4.95 | 4.67% |
| NH subtropics JJA | 1.38 | 2.92 | 30.5% | 6.10 | 6.05% |
| SH subtropics DJF | 1.37 | 2.85 | 30.3% | 6.10 | 5.77% |
| NH 50–70°N 120°W–120°E JJA | 1.37 | 2.85 | 31.1% | 6.23 | 6.20% |
| Southern Ocean DJF | 1.39 | 2.85 | 32.6% | 6.68 | 6.63% |
| Southern Ocean JJA | 1.63 | 3.59 | 38.1% | 9.89 | 8.70% |
| South America JJA | 2.46 | 5.19 | 53.6% | 25.4 | 16.1% |
| NH 50–70°N 120°W–120°E DJF | 1.61 | 3.51 | 37.6% | 9.66 | 8.47% |
| Scandinavia DJF | 1.76 | 3.84 | 41.0% | 11.8 | 10.4% |

the averaging effect. The corresponding noise-equivalent temperature amplitude is $\widehat{T}_{noise} = \sigma/\sqrt{n}$, i.e., about 0.13 K for both HIRDLS and SABER. According to Eq. (5) we can calculate the noise-equivalent gravity wave potential energy

$$E_{pot,noise} = \frac{1}{4}\left(\frac{g}{N}\right)^2\left(\frac{\widehat{T}_{noise}}{\overline{T}}\right)^2. \tag{8}$$

Inserting typical values for the lower stratosphere of $g$=9.8 m s$^{-2}$, $\overline{T}$=230 K, $N$=0.02 s$^{-1}$, and $\widehat{T}_{noise}$=0.13 K, we obtain $E_{pot,noise} \approx 0.02$ J kg$^{-1}$ for HIRDLS and SABER at altitudes around 30 km. Dividing this value by the respective medians given in Figs. 9 and 10, left column, and taking the base-10 logarithm, we obtain values between about -1.9 and -2.4 for all regions, except for mid and high latitude regions in the respective summer hemisphere (for these regions we obtain values of around -1). This means that in all PDFs shown in Figs. 9 and 10, right column, the peaks of the distribution, as well as a large





part of the tails of the distributions at low relative gravity wave potential energies are well resolved. It can be assumed that this finding also holds for the PDFs of gravity wave absolute momentum fluxes, and similar considerations can be made also for other altitudes.

## 5 Gravity wave intermittency investigated by the Gini coefficient

### 5.1 The Gini coefficient

While PDFs give comprehensive information on intermittency in a given time interval and region, it takes a large number of observations to create a robust PDF. This comes at the cost of losing spatial and temporal resolution, and, potentially, combining different regions of different intermittency into one PDF. In such cases, even the use of normalized values for creating a PDF does not help. Further, PDFs do not provide an integral number that allows to display the distribution of intermittency in global maps.

One way to overcome these limitations is the introduction of intermittency coefficients. For gravity wave absolute momentum fluxes observed by superpressure balloons Hertzog et al. (2008) introduced the Bernoulli coefficient, and the 90[th]-percentile coefficient. Later, however, Plougonven et al. (2013) found that for high resolution model simulations another coefficient, the Gini coefficient (Gini, 1912), would be preferable. Therefore, in the following, we will also use the Gini coefficient for displaying intermittency distributions. First, for calculating the Gini coefficient, the data set consisting of $N$ observations $f_i$ is 435 sorted, such that $1 \le i \le N$ with $f_i \le f_{i+1}$. After that, a set of cumulative sums $F_n$ is calculated for $n = 1, ..., N-1$:

$$F_n = \sum_{i=1}^{n} f_i \tag{9}$$

Then the Gini coefficient $I_g$ is defined as follows

$$I_g = \frac{\sum_{n=1}^{N-1} \left(n\overline{f} - F_n\right)}{\sum_{n=1}^{N-1} n\overline{f}} \tag{10}$$

with $\overline{f}$ the average of the data set.

Values of the Gini coefficient are between 0 and 1, and the Gini coefficient is the higher the stronger the intermittency. The two extreme cases are (1) all values $f_n$ are equal, and (2) all values are negligible (zero) except for one single value that dominates the average of the data set. In case (1), intermittency is as low as possible, and $I_g = 0$. In case (2), intermittency is as strong as possible, and $I_g = 1$.

One problem, however, remains. Because of the low number of available data points relatively large longitude/latitude bins 445 of $30°$ longitude $\times\ 20°$ latitude are required for determining global distributions of SABER absolute momentum fluxes. In the presence of strong spatial gradients of the global distribution, this will lead to biases of the global distribution of intermittency coefficients.

As can be seen from Eq. (10), the Gini coefficient is a relative measure and does not depend on the average magnitude of the data set. This means that, in order to reduce the biasing effect of spatial gradients within given longitude/latitude bins, similar





as for PDFs, single values can be normalized by, for example, the monthly average or median distribution. In regions of low gradients, normalization of values will leave values of $I_g$ unaltered, while in regions of strong gradients the use of normalized values will reduce biases of $I_g$. Considering a lognormal distribution, the median is the characteristic value of the PDF (see also Sect. 4). To reduce biases of the Gini coeefficient distributions we therefore apply the same procedure as for the normalized PDFs: Before calculating the global distributions of $I_g$, we generally normalize single values of gravity wave potential energies

or absolute momentum fluxes by the monthly global distribution of medians determined in the sets of longitude/latitude bins used in our study (cf. Sect. 3.2) and then interpolated to the location of each observation. This is performed separately for each month. It should be noted that global distributions of $I_g$ would be almost unchanged if we would normalize by the monthly global distribution of averages, instead of the monthly global distribution of medians.

     For HIRDLS, global distributions of $I_g$ are almost unchanged by this normalization procedure because relatively small

longitude/latitude bins of 15° longitude × 5° latitude are used for calculating global distributions. Obviously, the effect of spatial gradients within these small bins can be widely neglected. The same holds for global distributions of $I_g$ for SABER potential energies that are based on bins of 15° longitude × 10° latitude. For the SABER distributions of momentum fluxes, however, that are calculated with a bin size of 30° longitude × 20° latitude, values of $I_g$ in regions of strong spatial gradients are strongly reduced if normalization is applied, and the distributions become more similar to the corresponding HIRDLS

distributions of $I_g$. The resulting HIRDLS and SABER global distributions of Gini coefficients at 30 km altitude (i.e., at a relatively low altitude) will be discussed in the following subsection (Sect. 5.2).

## 5.2    Global distributions of Gini coefficients

Distributions of Gini coefficients were previously derived for gravity wave potential energies (e.g., Baumgaertner and McDonald, 2007), and for gravity wave absolute momentum fluxes (e.g., Hertzog et al., 2008, 2012; Wright et al., 2013). Calculation

of potential energies from satellite observations can be performed for single altitude profiles, while calculation of momentum fluxes requires assumptions how different altitude profiles can be combined. Particularly for SABER, the sampling distance for 50% of the observed altitude profiles is too large for momentum fluxes to be calculated. This means that the statistics for calculating global distributions is much better for potential energies. Further, for calculating momentum fluxes, a certain amount of altitude profiles has to be discarded due to non-matching vertical wavelengths (see also Ern et al., 2018, and references therein).

Therefore, global distributions of Gini coefficients for gravity wave potential energies are based on a much better statistics, and we will discuss these distributions first.

### 5.2.1    Gini coefficients for gravity wave potential energies

Figure 11 shows global distributions of Gini coefficients for HIRDLS gravity wave potential energies at 30 km altitude. The different panels in Fig. 11 represent different average calendar months. Averaging over the the respective monthly distributions

was performed for the period March 2005 until February 2008, i.e. three full years. For the global distributions, Gini coefficients were determined in overlapping bins of 15° longitude × 5° latitude, and only those bins were used that contain more than 40 data points. As discussed in Sect. 5.1, Gini coefficients were calculated from normalized potential energies, which means





that each potential energy data point was normalized by the $E_{pot}$ value of the corresponding monthly average median $E_{pot}$ distribution, interpolated to the exact location of each data point considered.

As can be seen from Fig. 11, the global distribution of Gini coefficients exhibits seasonal variations that are linked to seasonal variations of the potential energy distributions (see Fig. 1). Particularly high values of the Gini coefficient are found in the respective winter hemisphere at mid and high latitudes. In the winter season, these regions are dominated by the the polar night jets. The gravity waves in these regions are mainly excited by flow over orography (mountain waves), and by jet-related source processes. The corresponding gravity wave sources are highly variable, as they depend on the strongly variable

near-surface winds, and strongly variable wind jets and weather systems, respectively. In addition, the strong winds in the polar night jets offer favorable propagation conditions for the gravity waves excited by these mechanisms.

In the months April to October the latitude range of about 40°–65°S is dominated by the southern hemisphere polar night jet. During this period, maximum intermittency of $I_g$ up to values of about 0.7 is found over the southern tip of South America, and over the Antarctic Peninsula. These regions are well known as source regions of very strong and very intermittent mountain

waves (e.g., Hertzog et al., 2008). In the same period and latitude range, over the Southern Ocean intermittency is still quite strong with values of $I_g$ around 0.5.

Similarly, in the Northern Hemisphere the latitude range of about 40°–65°N is dominated by the northern hemisphere polar night jet during the months of November to February. Gini coefficients in the northern hemisphere polar night jet are about as strong as in the southern hemisphere polar night jet over the ocean. However, peak values as high as over the southern tip of

South America and the Antarctic Peninsula are not attained.

At latitudes equatorward of about 30° to 40°, Gini coefficients are comparably low (around 0.4). Still, there is a hemispheric asymmetry with somewhat higher values in the subtropics of the respective summer hemisphere. As can be seen in the global distributions of $E_{pot}$, there are enhancements in the summertime subtropics that are related to gravity waves that are excited by deep convection (e.g., Jiang et al., 2004; Wright and Gille, 2011; Ern and Preusse, 2012; Trinh et al., 2016; Kalisch et al., 2016;

Stephan et al., 2019b). These gravity waves seem to be somewhat more intermittent than convectively generated gravity waves in the tropics, which confirms our results obtained for the PDFs in Sect. 4.2.1. Also at midlatitudes in boreal summer over North America intermittency is somewhat enhanced, possibly related to thunderstorms in the summer season that are known to excite strong gravity waves (e.g., Hoffmann and Alexander, 2010).

Figure 12 shows the same as Fig. 11, but for the SABER instrument. However, for SABER, Gini coefficients were calculated

in overlapping bins of 15° longitude × 10° latitude because of the lower number of data points per month available for SABER. The SABER number of data points per month is particularly low at high latitudes during months when the SABER viewing geometry, and thus the covered latitude range changes. Another difference between Fig. 11 and Fig. 12 is that the distributions shown in Fig. 12 were obtained by averaging over a longer period (January 2002 until October 2020). The corresponding $E_{pot}$ distributions are shown in Fig. 2.

Obviously, the relative distributions of Gini coefficients in Figs. 11 and 12 are very similar, even though the spatial resolution of the SABER distributions is somewhat worse. On average, Gini coefficients for SABER $E_{pot}$ are somewhat higher. The reason for this effect is not known. Possibly, this effect is related to subtle differences in the HIRDLS and SABER sensitivity functions



**Figure 11.** Global distributions of Gini coefficients for HIRDLS gravity wave potential energies at 30 km altitude for each calendar month. Values shown are multi-year averages of monthly values determined in overlapping 15° longitude × 5° latitude grid boxes. Single $E_{pot}$ values were normalized by the monthly median distribution before calculating the Gini coefficients. Period used for averaging is March 2005 to February 2008.


**Figure 12.** Global distributions of Gini coefficients for SABER gravity wave potential energies at 30 km altitude for each calendar month. Values shown are multi-year averages of monthly values determined in overlapping 15° longitude × 10° latitude grid boxes. Single $E_{pot}$ values were normalized by the monthly median distribution before calculating the Gini coefficients. Period used for averaging is January 2002 to October 2020.





for detecting gravity waves. This indicates that the magnitude of the Gini coefficient somewhat depends on details of the dataset considered.

### 5.2.2 Gini coefficients for gravity wave absolute momentum fluxes

Figures 13 and 14 show global distributions of Gini coefficients $I_g$ for gravity wave absolute momentum fluxes for HIRDLS (Fig. 13) and SABER (Fig. 14). Again, multi-year average distributions were calculated for each calendar month. For HIRDLS, longitude/latitude bins were the same as for Fig. 11, but, as mentioned before, coarser longitude/latitude bins of 30° longitude × 20° latitude are used for the SABER Gini coefficients in Fig. 14. Again, bins of a given month and year were not considered

for calculating the multi-year averages if the bin contained fewer than 40 data points.

As expected, the relative distributions of Gini coefficients for momentum fluxes are very similar to those for potential energies (cf. Figs. 11 and 12). However, Gini coefficients for gravity wave absolute momentum fluxes are generally higher than those obtained for potential energies. Indeed, stronger intermittency for distributions of momentum fluxes would be expected. Different from potential energies, momentum fluxes also depend on gravity wave horizontal and vertical wavelengths, see

Eq. (7). As the wavelength distributions are also intermittent, this leads to the observed stronger intermittency of momentum fluxes. This effect was also seen for the PDFs in Sect. 4.2.

The relative distributions of the HIRDLS Gini coefficients for single calendar months in Fig. 13 are very similar to those previously derived by Wright et al. (2013) for HIRDLS absolute momentum fluxes (see the supplement of their paper). However, our Gini coefficients are considerably higher. Even the Gini coefficients for our gravity wave potential energies exceed the

Gini coefficients for gravity wave absolute momentum fluxes in Wright et al. (2013). One possible reason is the larger range of vertical wavelengths covered by our gravity wave analysis: vertical wavelengths of up to 25 km in our analysis compared to only up to 16 km in the analysis by Wright et al. (2013). As gravity wave momentum flux is proportional to the vertical wavelength (see Eq. (7)), our analysis might cover a larger number of high momentum flux events. In addition, the method described in Sect. 3 and Ern et al. (2018) provides a fixed vertical resolution of 10 km, resulting in good vertical resolution

also for long vertical wavelength gravity waves, and thus for events of potentially large momentum flux. Further, the method for deriving absolute momentum fluxes from pairs of altitude profiles in Wright et al. (2013) is somewhat different from the method used here. While pairs of altitude profiles with non-matching vertical wavelength are omitted in our study, these events are kept in Wright et al. (2013) and may result in a population of relatively small absolute momentum fluxes.

Remarkably, the magnitude of the Gini coefficients in Figs. 13 and 14 is very similar to values derived for absolute momen-

tum fluxes obtained from superpressure balloons and high resolution model simulations. For example, at altitudes of around 18 km Gini coefficients $I_g$ of 0.8 were obtained by Plougonven et al. (2013) over mountainous terrain in the Southern Hemisphere for the period September 2005 until February 2006 in simulations of the Weather Research and Forecast model (WRF). Similarly, a value of $I_g$=0.73 was obtained by Jewtoukoff et al. (2015) over mountainous terrain in the Southern Hemisphere during October 2010 from superpressure balloon observations. Over flat terrain (mostly ocean), values of $I_g$=0.34 to 0.58

($I_g$=0.44 on average) and $I_g$=0.36 to 0.51 were obtained, respectively, by the same studies. In the tropics, values of $I_g$ between





**Figure 13.** Same as Fig. 11, but Gini coefficients for HIRDLS gravity wave absolute momentum fluxes.





**Figure 14.** Same as Fig. 12, but Gini coefficients for SABER gravity wave absolute momentum fluxes and using a larger bin size of 30° longitude × 20° latitude.





0.48 and 0.59 were obtained from superpressure balloon observations by Jewtoukoff et al. (2013). (These tropical values might, however, be somewhat high-biased (cf. Corcos et al., 2021).)

The above mentioned values obtained from simulations and superpressure balloon observations compare very well to the values we obtain in our analysis from satellite data at 30 km altitude. For October, on average, we obtain values of about $I_g$=0.7
(for HIRDLS) and $I_g$=0.65 (for SABER) over the southern tip of South America and the Antarctic Peninsula (cf. Figs. 13j and 14j). In the tropics, we find average values between about 0.45 and 0.55, see also Figs. 13 and 14.

## 6 Vertical evolution of intermittency diagnosed by Gini coefficients

Global distributions of gravity wave intermittency in the lower stratosphere still contain much information about the intermittency introduced by the gravity wave source processes. However, also the vertical evolution of gravity wave intermittency is
of relevance, as the interaction of gravity waves with the background atmosphere during their propagation will have effect on the intermittency distribution. For example, while propagating upward, large-amplitude gravity waves will more easily saturate and dissipate than low-amplitude gravity waves. As high-amplitude gravity waves are usually located in the long tail of the PDFs at high values of momentum fluxes or potential energies, this effect should reduce the observed intermittency at high altitudes. However, also physical processes, like generation of secondary gravity waves when a primary gravity wave dissipates
(e.g., Vadas et al., 2018; Becker and Vadas, 2020), can alter intermittency and lead to variations of the observed intermittency distribution. Further, temporal changes of the background atmosphere will alter the propagation conditions for gravity waves, and can therefore introduce additional intermittency. Vertical changes of intermittency can therefore also serve as a benchmark for the quality of the global distribution of gravity waves in models, i.e. whether sources, gravity wave propagation conditions, and all relevant processes of gravity wave physics are realistically simulated.

### 6.1 Zonal average cross sections of Gini coefficients

In a first step, we investigate zonal average distributions of median gravity wave momentum fluxes, and of zonal average Gini coefficients. These distributions are shown for SABER in Fig. 15 for medians of absolute momentum fluxes, and for the corresponding zonal average Gini coefficients in Fig. 16.

For solstice conditions, enhanced gravity wave momentum fluxes are found in the polar night jets, as well as in the subtropics
of the respective summer hemisphere. Correspondingly, Gini coefficients are enhanced in the polar night jets. However, in the summer hemisphere highest Gini coefficients are found somewhat poleward of the momentum flux maximum, possibly because gravity waves are excited by deep convection more continuously in the tropics and subtropics, but more sporadically at midlatitudes.

Interestingly, the altitude dependence of Gini coefficients displays characteristic latitudinal differences. In the tropics, Gini
coefficients at low altitudes are generally low, and they increase with altitude. At low altitudes Gini coefficients should still be dominated by the intermittency of the gravity wave source processes. This indicates that in the tropics gravity waves of horizontal wavelengths longer than 100–200 km that can be detected by limb sounders are continuously excited (for example,



**Figure 15.** Multi-year average zonal average distributions of SABER median absolute gravity wave (GW) momentum fluxes for each calendar month. Averaging period is January 2002 until October 2020. Overplotted contour lines are zonal winds taken from the SPARC climatology of zonal winds (Swinbank and Ortland, 2003; Randel et al., 2002, 2004).



**Figure 16.** Same as Fig. 15, but for zonal average Gini coefficients of the absolute momentum flux distributions.





by mesoscale convective systems), and exceedingly strong events are rare, resulting in relatively low intermittency. This was already seen from the low-latitude PDFs in Figs. 7 and 9.

With increasing altitude, the intermittency of gravity waves, and thus the Gini coefficients, should be more and more dominated by the gravity wave propagation conditions that are governed by variations of the background atmosphere, i.e. the background winds and static stability. The background atmosphere in the tropical stratosphere is dominated by the relatively slow quasi-biennial oscillation of the zonal winds. Of course, the QBO will modulate the distribution of gravity waves (e.g., Ern et al., 2014). However, during a single month (our multi-year averages are based on single-month distributions) these vari-
ations will not add much intermittency to the intermittency caused by the gravity wave source processes. At higher altitudes, however, the background atmosphere in the tropics is dominated by the semiannual oscillation (SAO) of the zonal wind, as well as by atmospheric tides. The time scales of these variations are much shorter than that of the QBO, and, accordingly, on a monthly basis, these variations and the corresponding variations of the gravity wave distribution (e.g., Preusse et al., 2001; Ribstein and Achatz, 2016; Ern et al., 2015, 2021) should add intermittency to the monthly averages. This could explain the
increase of intermittency with altitude in the tropics, as well as the intermittency maxima that are found at low latitudes around 80 km altitude around the equinoxes (cf. Fig. 16). This will be investigated in more detail in Sect. 6.2.

In the respective winter hemisphere, at low altitudes, we find enhanced Gini coefficients in the region of the polar night jet. As was seen from Figs. 1–4 and Figs. 11–14, enhanced gravity wave activity and Gini coefficients in this season and these regions are likely caused by orography and jet-related gravity wave sources. Strikingly, Gini coefficients stay high until
reaching about the altitude of maximum eastward winds. Above this altitude, Gini coefficients decrease considerably. One possible explanation for this finding could be that high amplitude mountain waves, but also high amplitude jet-generated waves, will gradually saturate due to amplitude growth in the reduced background density and dissipate while propagating upward (see also Alexander et al., 2016). Since these waves contribute considerably to increased intermittency at low altitudes, dissipation of these waves should lead to a decrease of Gini coefficients with altitude. This was already indicated by the SABER
PDFs at different altitudes shown in Fig. 6. Consequently, gravity wave hotspots should become more and more less prominent at high altitudes. Again, this will be investigated in more detail in Sect. 6.2.

In the respective summer hemisphere, mid-latitude Gini coefficients are somewhat higher than in the tropics, but peak values are considerably lower than peak values found in the in the polar night jets. There is some altitude variation with somewhat enhanced values at low altitudes, possibly caused by intermittent gravity wave sources. Further, Gini coefficients are somewhat
enhanced in the region of maximum westward winds in the summertime mesospheric wind jets, which hints at some effect of changes in gravity wave propagation conditions.

## 6.2   Horizontal distributions at different altitudes

Next, we will investigate how the global distributions of gravity wave momentum fluxes and Gini coefficients evolve with increasing altitude. We will focus on SABER observations because these are available over a larger altitude range. Further, we
focus on periods around the solstices in order to investigate the changes in the very characteristic global distributions during these periods. In addition, we want to avoid the months of TIMED yaw maneuvers. For these reasons, we selected the calendar





months of August and December: Fig. 17 shows for the calendar month of August the global distributions of SABER median gravity wave momentum fluxes (left column) and the corresponding distributions of Gini coefficients (right column) at altitudes of 30 km (upper row), 50 km (second row), 70 km (third row), and 80 km (bottom row). Figure 18 shows the same, but for the calendar month of December. Again, the distributions shown are averages over the years 2002 until 2020.


As can be see from Figs. 17 and 18, at an altitude of 30 km the global distributions of momentum fluxes and Gini coefficients show the characteristic features of enhanced momentum fluxes and Gini coefficients at mid and high latitudes of the respective winter hemisphere (that are governed by the respective polar night jet), low Gini coefficients in the tropics, and intermediate Gini coefficients in the summer hemisphere subtropics and midlatitudes. Particular enhancements of momentum fluxes and

Gini coefficients are found in August over South America and the Antarctic Peninsula, and in December enhancements are found over Northeast America, Europe, and Northeast Asia. Further, enhancements of momentum fluxes are found in the subtropics of the respective summer hemisphere.

At an altitude of 50 km, these distributions are still visible, but the specific features are much less pronounced. At 70 km altitude, the momentum flux distributions exhibit still some of these characteristic features. The range of the logarithmic

color scale, however, was much reduced to make these structures visible. Different from this, the global distribution of Gini coefficients is almost flat and attains intermediate values of about 0.5, which means an increase of Gini coefficients in the tropics, and a reduction of Gini coefficients at mid and high latitudes in the respective winter hemisphere.

For December, this is also the case at 80 km altitude, while in August at 80 km altitude some structures start to emerge, with enhanced Gini coefficients in the tropics, and enhanced values at midlatitudes in the Northern Hemisphere. The seasonality

of these structures might be related to seasonal variations of the background atmosphere, which can lead to modulations of the gravity wave distribution. Possible candidates are the SAO in the tropics, and atmospheric tides. Further investigation of this effect is, however, beyond the scope of this study. At 80 km also the momentum flux distribution shows some structure. However, it is not clear whether the strong enhancement of momentum fluxes at high southern latitudes during August is reliable, as enhanced noise of the SABER temperature retrieval is expected in the cold summer mesopause region (see also Ern

et al., 2018).

Changes of the global distribution of Gini coefficients with altitude were investigated previously, for example, by Alexander et al. (2016) based on gravity wave momentum fluxes of the gravity waves resolved by the Kanto GCM in the Southern Hemisphere. The Kanto model resolves gravity waves of horizontal wavelength longer than about 200 km (e.g., Watanabe et al., 2008; Sato et al., 2012), which roughly agrees with the range of horizontal wavelengths seen by infrared limb sounders. In

July, Alexander et al. (2016) found at 50 hPa (about 21 km altitude) that Gini coefficients at mid and high southern latitudes are about 0.55 for oceanic regions, while in the regions above South America and the Antarctic Peninsula (that are dominated by mountain waves) Gini coefficients reach peak values as high as 0.7 to 0.9. At higher altitudes of 0.1 hPa (about 65 km), this difference between regions is strongly reduced to about 0.55 to 0.6 for the oceanic regions and 0.6 to 0.65 for the mountain wave regions, respectively (see their Figs. 5b and 6). In the subtropics, Gini coefficients increase from about 0.45 to 0.50 at

50 hPa to about 0.55 at 0.1 hPa.

**Figure 17.** Global distributions of **(a, c, e, g)** SABER median absolute gravity wave momentum fluxes (MF) at altitudes of 30, 50, 70, and 80 km for the calendar month of August (left column), as well as **(b, d, f, h)** the corresponding global distributions of Gini coefficients. Again, the global distributions are averages obtained over the period January 2002 until October 2020.



**(a) SABER MF z=30km**

**(b) SABER Gini z=30km**

**(c) SABER MF z=50km**

**(d) SABER Gini z=50km**

**(e) SABER MF z=70km**

**(f) SABER Gini z=70km**

**(g) SABER MF z=80km**

**(h) SABER Gini z=80km**

**Figure 18.** Same as Fig. 17, but for the calendar month of December.





In January, the Gini coefficients obtained by the Kanto model are about 0.45 to 0.50, throughout, in the Southern Hemisphere at 21 km altitude. Gini coefficients increase only slightly between 21 km (50 hPa) and 65 km altitude (0.1 hPa), see Figs. 5a and 10a in Alexander et al. (2016).

Qualitatively, these results are similar to our findings: an increase of Gini coefficients with altitude at low latitudes, and a
more and more uniform distribution of Gini coefficients with increasing altitude. Even the magnitudes of Gini coefficients are in good agreement between Alexander et al. (2016) and our study.

Changes of Gini coefficients with altitude were also investigated by Wright et al. (2013) using HIRDLS absolute momentum fluxes. Of course, as mentioned in Sect. 5.2.2, there is some difference in the magnitude of Gini coefficients between Wright et al. (2013) and our study. However, there is still qualitative agreement in some of the relative variations. For example, in the
tropics Wright et al. (2013) observed an increase of Gini coefficients with altitude at altitudes above about ∼40 km, similar as in our study. In other regions the agreement with our study is less clear. For example, in Wright et al. (2013) intermittency increases or stays constant with altitude in oceanic regions, and either slightly increases or decreases with altitude in mountain wave regions.

## 7 Summary and discussion

Intermittency of the global distribution of gravity waves is an important effect that can be caused, for example by gravity wave source processes, or varying gravity wave propagation conditions, i.e. variability of the background atmosphere. Although intermittency has effect on where gravity waves dissipate and, thereby, contribute to the driving of the global atmospheric circulation, intermittency is often neglected in gravity wave parameterizations. This means that the variability of gravity wave drag that is introduced by gravity wave intermittency is missing in many models.

As there are only few observations of gravity wave intermittency, and of its variation with altitude, we used global observations of gravity wave potential energies, and of gravity wave absolute momentum fluxes, by the infrared limb sounding satellite instruments High Resolution Dynamics Limb Sounder (HIRDLS) and Sounding of the Atmosphere using Broadband Emission Radiometry (SABER) to investigate the global distribution of gravity wave intermittency and its evolution with altitude. For HIRDLS we use observations from the period January 2005 to February 2008, and for SABER from the period January 2002
until October 2020. The determination of gravity wave potential energies and absolute momentum fluxes from temperature observations follows the method described, for example, in Ern et al. (2018).

First, we derived probability density functions (PDFs) of gravity wave potential energies and absolute momentum fluxes in different regions. For a better comparability between different regions, we normalized the single values of potential energies and momentum fluxes by their monthly global distribution of median values. In agreement with previous observations (e.g.,
Baumgaertner and McDonald, 2007; Hertzog et al., 2008, 2012; Plougonven et al., 2013; Wright et al., 2013), we found that the PDFs of gravity wave potential energies and momentum fluxes roughly follow lognormal distributions. Already this fact is an indication for strong intermittency associated with the distribution of gravity waves. However, there are important differences, depending on region and altitude.





In the stratosphere, we find that the PDFs over the Southern Ocean during austral winter closely follow lognormal distributions, in agreement with previous observations by HIRDLS and superpressure balloons (Hertzog et al., 2012). Our satellite observations show that the same is also the case at northern mid and high latitudes during boreal winter.

From previous observations by superpressure balloons and from corresponding high resolution model simulations in Southern Hemisphere regions that are dominated by mountain waves during austral winter, it is expected that PDFs of absolute momentum fluxes even exceed a lognormal distribution at high momentum flux values (e.g., Hertzog et al., 2012; Plougonven et al., 2013). This is also found in our satellite observations, but less pronounced, which is likely an observational filter effect: satellite infrared limb sounders are sensitive only to gravity waves of horizontal wavelengths $>100$–$200\,\mathrm{km}$ (e.g., Preusse et al., 2002; Ern et al., 2018), and might therefore miss events of very strong momentum fluxes. Further, low-biases of momentum fluxes may be introduced by (1) the HIRDLS and SABER sensitivity functions for gravity waves, (2) the satellite sampling that allows only to estimate the apparent gravity wave horizontal wavelength parallel to the satellite track, which always overestimates the true gravity wave horizontal wavelength, and (3) short horizontal wavelength gravity waves may be undersampled, which also leads to an overestimation of the true gravity wave horizontal wavelength. A more detailed discussion of the observational filter of infrared limb sounders is given, for example by Preusse et al. (2009a) and Trinh et al. (2015).

The PDFs over Scandinavia during boreal winter do not show this pronounced enhancement at high momentum flux values. Possibly, the gravity waves observed in this region are a mixture of highly intermittent mountain waves and less intermittent jet-generated gravity waves.

The PDFs in the tropics and in the respective summer hemisphere are quite different from those observed at mid and high latitudes during winter. The tropical and summer hemisphere PDFs are considerably skewed toward low values: low momentum flux values are more abundant, and high momentum fluxes are less abundant than suggested by a lognormal distribution. This is in agreement with previous findings by, for example, Ern et al. (2014), and this finding may be relevant for a more realistic simulation of the QBO.

At high altitudes, the PDFs are more and more skewed towards low values because the tail of the distribution at high momentum flux values is much reduced, likely due to dissipation of high-amplitude gravity waves at already relatively low altitudes.

The PDFs we find for gravity wave potential energies in the same regions are qualitatively very similar to those found for absolute momentum fluxes. The main difference is that potential energy PDFs are much narrower. Indeed, the intermittency of potential energies should be lower than the intermittency of momentum fluxes because in the calculation of momentum fluxes also the gravity wave horizontal and vertical wavelengths are entering, which will introduce further variability and thus intermittency.

While PDFs can give a complete picture of the magnitude distribution of a selected parameter, they require a large number of observations to give reliable results, and a display of global distributions is not easily possible. For this reason, intermittency coefficients were developed that provide a measure of intermittency for a given data set as a single number (see also Hertzog et al., 2012; Plougonven et al., 2013). Recently, the Gini coefficient (Gini, 1912) has most widely been used to characterize the intermittency of gravity wave distributions. Therefore, we derived global distributions of Gini coefficients for each multi-year





average calendar month for both HIRDLS and SABER gravity wave potential energies and absolute momentum fluxes. Again,

single values of potential energies and momentum fluxes are normalized by their monthly median distributions to avoid biases of derived Gini coefficients by horizontal gradients within the longitude/latitude bins used to calculate the Gini coefficients. This is particularly important for SABER momentum fluxes because relatively large bins of 30° longitude × 20° latitude have to be used to obtain sufficient statistics.

As already indicated by the PDFs, intermittency in the stratosphere is weakest in the tropics, followed by the summertime

subtropics and summertime midlatitudes. Intermittency is strongest at winter hemisphere mid and high latitudes, with the strongest values over regions that are known for the strong activity of mountain waves, for example South America, the Antarctic Peninsula, New Zealand, or Scandinavia. The magnitude of our Gini coefficients is in good agreement with previous observations by superpressure balloons (e.g., Plougonven et al., 2013; Jewtoukoff et al., 2013, 2015), and with previous results obtained with high resolution models (e.g., Plougonven et al., 2013; Alexander et al., 2016). Values of Gini coefficients obtained

previously from HIRDLS observations are considerably lower (see the supplement of Wright et al. (2013)). Possible reasons are differences in the method for deriving gravity wave momentum fluxes, as well as a larger range of vertical wavelengths covered in our work.

In the extratropics, where intermittency is strongest at low altitudes, intermittency decreases with altitude. In the tropics, however, intermittency increases with altitude. Consequently, the global distribution of Gini coefficients is relatively flat at

altitudes around 70 km. These findings are in good agreement with results of the Kanto model in the Southern Hemisphere (Alexander et al., 2016). In the tropics, our findings are also qualitatively in agreement with previous results obtained from HIRDLS observations (Wright et al., 2013).

In regions of strong intermittency at low altitudes, the reduction of Gini coefficients with altitude may be related to the dissipation of high amplitude gravity waves, as these waves saturate more quickly while propagating upward. Increases of

Gini coefficients with altitude (particularly in the tropics) may be related to varying gravity wave propagation conditions. In the tropics, this may be caused, for example, by the semiannual oscillation (SAO) of the background winds, as well as by atmospheric tides (e.g., Ern et al., 2015, 2021; Ribstein and Achatz, 2016), as the time scales of these variations are close to, or shorter, than one month, which is the time interval used in our study to collect observations for calculating Gini coefficients.

It should be mentioned that the results obtained in our study apply only for the part of the gravity wave spectrum that

is visible for infrared limb sounders. Our study roughly covers gravity waves of horizontal wavelengths longer than about 100–200 km, and vertical wavelengths in the range 2–25 km for HIRDLS, and 4–25 km for SABER. Approximate sensitivity functions are given, for example, in Ern et al. (2018). An improvement of this sensitivity could be achieved, for example, by future 3D observations and tomographic temperature retrievals (e.g., Preusse et al., 2009a; Ungermann et al., 2010; Song et al., 2017, 2018; Gumbel et al., 2020). It should be noted that the level of intermittency depends on the part of the gravity

wave spectrum considered. In particular, intermittency could be stronger if only short horizontal wavelength gravity waves are considered. This is indicated, for example, in the recent studies by Kim et al. (2021), or Corcos et al. (2021). Further, intermittency could be different if time scales much shorter than one month are used to collect data for calculating PDFs, or Gini coefficients.



*Data availability.* The satellite data used in our study are open-access data: SABER data are available from GATS Inc. at http://saber.

gats-inc.com/browse_data.php (last access: 19 May 2022, GATS Inc., 2022). HIRDLS level-2 data are available via the NASA Goddard

Earth Sciences Data and Information Services Center (GES DISC) at https://acdisc.gesdisc.eosdis.nasa.gov/data/Aura_HIRDLS_Level2/

(last access: 19 May 2022, NASA GES DISC, 2022)

The SPARC temperature and zonal wind climatology (http://www.sparc-climate.org/data-center/data-access/reference-climatologies/randels-climatologies/

temperature-wind-climatology/) is available at ftp://sparc-ftp.ceda.ac.uk/sparc/ref_clim/randel/temp_wind/ (last access: 19 May 2022, SPARC,

760 2002).

*Author contributions.* ME designed and performed the technical analysis. PP, and MR contributed with ideas to the scientific interpretation

of results. All coauthors contributed to the interpretation of results and preparation of the paper.

*Competing interests.* The authors declare that they have no conflict of interest.

*Acknowledgements.* We would like to thank the teams of the HIRDLS and SABER instruments, as well as the SPARC data center for

creating and maintaining the excellent data sets used in our study. ME would like to acknowledge the fruitful discussions with the Gravity

Wave Group at the International Space Science Institute (ISSI), Bern.

*Financial support.* This work was supported by the Deutsche Forschungsgemeinschaft (DFG, German Research Foundation) projects ER 474/4–

2 and PR 919/4–2 (MS–GWaves/SV), which are part of the DFG research unit FOR 1898 (MS–GWaves). This work was also supported by

the Federal German Ministry for Education and Research (Bundesministerium für Bildung und Forschung, BMBF) project QUBICC, grant

770 number 01LG1905C, which is part of the Role of the Middle Atmosphere in Climate II (ROMIC-II) program of BMBF.



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
