# Peer review of "Intermittency of gravity wave potential energies and absolute momentum fluxes derived from infrared limb sounding satellite observations"

_Atmospheric Chemistry and Physics, 2022_

## Author Comment (AC2)

Many thanks to Referee #1 (Corwin Wright) for appreciating our work and the very helpful comments that will significantly help to improve the manuscript!

Please find below our point-by-point reply to the reviewer concerns. Comments by Reviewer #1 are given in red, our reply is given in black, and changes in the manuscript are indicated in blue.

**Reply to the Main Concerns by Reviewer # 1:**

(Main Comment 1:) I do have a probable answer to one question raised in the manuscript. In several places (e.g lines 532-543, lines 657-664, line 730) the authors highlight that their results appear quantitatively inconsistent with a previous study (W2013a, reference below). Specifically, the wave intermittencies measured in the current study are consistently higher than those seen in W2013a. However, I believe this arises from a key methodological difference. In W2013a, we used an approach (described by WG2013, reference below) which selects for multiple waves in a given HIRDLS measurement, and which typically (see W2015, reference below) identifies four discrete waves at any one measurement location. In contrast, the current study identifies at most a single wave at each point (L173). These 'additional' waves in W2013a tend to be lower-amplitude and have smaller momentum fluxes (WG2013, W2015) than the 'main' waves measured by the method used here, and will hence tend to strongly pull intermittency values down. I think this methodological difference is likely to explain most if not all of the differences between the current study and W2013a.

I emphasise that this difference is not an mistake in experimental design in the current manuscript and I have no objections to the authors using a more cautious approach and identifying at most one wave as they do here - it is a perfectly valid choice. The WG2013 method has the advantage that it detects more smaller waves from the same data, but in particularly for HIRDLS can be negatively affected by a known fault with the instrument which will introduce a small height-varying population of nonexistent waves into the data (W2015) and hence would have to be treated cautiously for a study of this type. We had not identified this problem at the time of writing W2013a, but it shouldn't affect the results presented there too much as the effect is very small at the low altitudes that study focuses on, and even at high altitudes is only a few percent of the measured waves - i.e. I suspect that the differences between the current study and W2013a will be almost entirely methodological rather than due to this data issue. [W2013a] Wright, Osprey and Gille, JGR 2013: 10.1002/jgrd.50869 [WG2013] Wright and Gille, GRL 2013: 10.1002/grl.50378 [W2015] Wright, Osprey and Gille, ACP 2015:

10.5194/acp-15-8459-2015

Thank you very much for this clarification! In our manuscript we already suggested that differences might occur due to a population of small-amplitude waves that lead to a different shape of the PDFs and reduce intermittency, however we did not know the exact reason for these low amplitude waves.

We will add this explanation in the revised manuscript after former I. 532 when the difference in magnitude is first mentioned and potential reasons are given.

"The likely main reason for this difference in magnitude are differences in the gravity wave analysis technique. While in our study we focus on only the strongest gravity wave at a given altitude, the method used by Wright et al. (2013) selects for multiple waves in a given HIRDLS measurement, and which typically identifies four discrete waves at any one measurement location (Wright and Gille, 2013; Wright et al., 2015). These additional waves usually have lower amplitudes and carry small momentum fluxes. This large population of relatively small

absolute momentum fluxes will considerably pull down the level of intermittency, while relative variations of intermittency should be still dominated by the largest events."

In addition, we have shortened the part summarizing other potential reasons for differences in magnitude.

Further, we will mention in the summary that a direct comparison of intermittency is only possible if similar analysis methods are used.

(Main Comment 2) The methodological choice to normalise the distributions (L238-247) does make sense when the results are considered, but probably needs a little more justification. For QBO regions, where filtering varies strongly from year to year, the logic is clear and coherent, but I am less clear on the justification for doing so at extratropical latitudes (at least outside SSW periods). If this was a minor point I would be happy with the current presentation, but since this choice underpins most of the results presented I think it needs to be justified a bit more strongly.

There are several reasons for normalizing the distributions:

- Different parameters can be directly compared using the same scales
- Normalizing distributions accounts for spatial gradients within the area considered. These spatial gradients can introduce spurious intermittency. This is particularly important for PDFs that are created from quite large regions, but also for global distributions that result from gridding using a set of lon/lat bins. For example, for creating global distributions of SABER gravity wave momentum fluxes we require relatively large lon/lat-bins for averaging because of the sparsity of data. In an appendix we now show the global distributions of Gini coefficients for SABER momentum fluxes without normalizing the distributions beforehand. These distributions are clearly biased due to spatial gradients within the bins. This effect is strongest at southern hemisphere mid to high latitudes during austral winter.
- Normalizing the distributions generally reduces the width of PDFs (intermittency) in all cases, which shows that normalization is generally beneficial.

In the revised manuscript, we have given further reasoning after former I.240 why we apply normalization. This is followed by a detailed description of the normalization procedure. Further, we have added in an appendix the distributions of Gini coefficients for SABER gravity wave absolute momentum fluxes without normalization being applied. These distributions clearly show spurious enhancements of intermittency in regions of strong spatial gradients.

(Main Comment 3) In section 5.1, I was still confused after several readings as to exactly how the gradient effect (line 444 onwards) was being compensated for - if the normalisation is taking place at the level of the bin, then how does this reduce spatial biasing due to gradients within the bin? I suspect I am misunderstanding something here, and as such would appreciate this section being made clearer.

The problem is that by forming a PDF (or quantifying intermittency in another way) one assumes that all data points considered follow the same distribution with the same mean and the same standard deviation. This, however, is clearly not the case if there are horizontal gradients caused by variations of the overall global distribution within an area considered. These variations are compensated for by normalizing the single values by the temporally and spatially varying global distribution of medians.

This reasoning will be given in the revised manuscript after former I.240.

(Main Comment 4) A minor concern I do have is that the manuscript feels very long and could probably benefit from trimming, but this is not a critical problem and the paper does contain a lot of data which does justify this. If the authors do choose to trim it, I think it would be better to do so by slimming down each section rather than removing some parts entirely, and by reducing repetition between sections.

As recommended, the manuscript has been shortened in some places.

**Reply to the Additional Comments by Reviewer # 1:**

(1) L131: the resolution HIRDLS drops sharply note that of above 60km. 2km Figaveraging above this level - see e.g. HIRDLS HIRDLS 5.1.1 the Data Quality Document DQD: ure of https://docserver.gesdisc.eosdis.nasa.gov/repository/Mission/HIRDLS/3.3\_Product\_Documentation/3.3. DQD\_V7.pdf

This will be mentioned in the revised paper by adding:

"( $\sim$ 2 km above 60 km altitude)"

**(2) L148: what kind of high-pass filtering is being applied here? Some types could introduce small waves, which may pull measured intermittencies down.**

The high-pass filtering applied here is just fitting in the vertical a single sinusoidal wave of vertical wavelength between 40 and 80km. Since this vertical wavelength is relatively long, its variations in the vertical are very moderate within the 10km vertical intervals we are using for fitting wave amplitudes.

We will add the following information for clarification.

"This high-pass is performed by fitting and subtracting a sinusoidal wave of vertical wavelength of 40 km or longer, individually for each altitude profile."

(3) L201 and other places: while it's reasonable clear that by 'average' the authors mean 'mean', the mode, median and mean are all types of average - I would suggest switching from 'average' to 'mean' throughout to avoid any confusion.

Thank you very much for this pointing this out! For clarification, we have changed the text in former I. 201 to:

"... values are multi-year means of medians, and not multi-year means of arithmetic mean values."

Further, we now use 'mean' instead of 'average', where applicable.

(4) L273: additionally, presumably SABER cannot access the smallest horizontal wavelengths due to the inter-profile spacing being about double that of HIRDLS. This is mentioned in s4.2.2 later, but this is the first mention where that effect is relevant so it might help to put it here instead.

This effect was already mentioned in former lines 274–276. For better clarification we have added:

"(about twice the along-track sampling step of HIRDLS)"

(5) L339 onwards: similar high intermittencies are seen in the open Southern Ocean using AIRS data in Hindley et al 2019 (their figure 9). While this has a very different observational filter, it may be relevant to the discussion here too. Hindley et al, ACP 2019: doi:10.5194/acp-19-15377-2019

Indeed, it is quite remarkable that the intermittency is very similar although the instruments and their observational filters are very different! We have added another paragraph after former I. 355 that mentions the AIRS results by Hindley et al. (2019).

"It is also noteworthy that gravity wave observations by the Atmospheric Infrared Sounder (AIRS) show similar characteristics at mid and high southern latitudes (Hindley et al., 2019). Although AIRS has a very different observational filter and observes only gravity waves of vertical wavelength longer than about 12 km (see, for example, Ern et al., 2017; Meyer et al., 2018), very strong intermittency is found over South America and the Antarctic Peninsula, and somewhat weaker, but still strong, intermittency over the Southern Ocean."

**(6) Section 4.2.4 - I like this!**

Thank you very much for appreciating our work!

(7) L470: I understand here that you mean individual profiles (i.e. single profiles against altitude), but the text as-written could be read as referring to single levels \*within\* a profile (i.e. a single altitude level of a given profile) - it might help to clarify this.

Thank you very much for pointing this out! We now use the expression 'for each individual altitude profile'.

"... Based on gravity wave amplitudes, calculation of potential energies from satellite observations can be performed for each individual altitude profile, ..."

**(8) L484: This normalisation depends on the averaging method used and more detail about this would help - for example, are the averaged values being stored at the grid-corner or grid-centre before being interpolated to the profile location?**

As averaging method we selected the median, which is quite convenient: if the single values follow a lognormal distribution, the PDFs of the normalized distribution will be centered at zero on a log scale. The median values are stored at the grid centers, and linear interpolation in longitude and latitude is applied to obtain an interpolated value at the profile locations.

In the revised manuscript, we will add a detailed description of the normalization procedure after former I.240. See also Minor Comment (10) by Reviewer # 2, and Minor Comment 2 by Reviewer # 3.

**(9) L515 onwards: could the lack of shorter vertical wavelengths and/or different normalisation areas cause this higher Gini coefficient estimate?**

Indeed, the SABER lack of shorter vertical wavelengths could be a candidate for differences between SABER and HIRDLS Gini coefficients. Such effects were already covered in the manuscript by stating: "... differences in the HIRDLS and SABER sensitivity functions for detecting gravity waves."

The size of the normalization areas should not be relevant, because for  $E_{pot}$  we use relatively small lon/lat bins for both instruments.

In addition to these points, in reply to Reviewer # 3, Minor Comment # 1, we also added some discussion on potential effects of the different line of sight orientations.

**(10) L544: this is remarkable given the very different observational filters involved - do you think this is a real similarity and (e.g.) that the same intermittency effects are being observed uniformly across the GW spectrum despite the very different physical scales, do you think the methods are actually observing the same waves, or do you think it's just a coincidence?**

It looks like a real similarity, at least for a certain part of the gravity wave spectrum. We have added this speculation after former I.556:

"Because the observational filters of limb sounders and superpressure balloons are very different, and also the gravity wave spectrum in the simulations will be different, not necessarily the same waves are being observed. Therefore, the agreement in Gini coefficient magnitudes suggests that, at least over a certain part of the gravity wave spectrum, the statistical distributions of momentum fluxes are similar."

---

## Author Comment (AC3)

We would like to thank Referee #2 for appreciating our efforts and for the very helpful comments and suggestions that will definitively improve the manuscript!

Please find below our point-by-point reply to the reviewer concerns. Comments by Reviewer #2 are given in red, our reply is given in black, and changes in the manuscript are indicated in blue.

**Reply to the Minor Comments by Reviewer # 2:**

**(Minor Comment 1) L3: 'effect' of what ?**

For clarification, this sentence has been reworded to: "One important, but often neglected characteristic of the gravity wave distribution is the fact that..."

**(Minor Comment 2) Eqs. (4)-(5): Would this part be fit more in Section 3.2 than here ?**

As recommended, Eqs. (4) and (5) have been moved to Section 3.2.

**(Minor Comment 3) L122: 'every about 60 days for about 60 days': Would the second part be removable ?**

Removed as recommended.

**(Minor Comment 4) L134: One of the two 'instrument' should be removed.**

Removed as recommended.

**(Minor Comment 5) L146: 'location': Would it be replaced with 'latitude, height' ? (if I understand correcity)**

Here the "location" includes not only latitude and height, but also longitude. This has now been clarified in the revised manuscript as follows: location (i.e., longitude, latitude, and height)

**(Minor Comment 6) L156: 'different': between what ?**

The sentence has been reworded for clarification:

"The reason is that during these short time intervals the respective local solar times of the ascending and descending parts of the satellite orbit are about constant."

**(Minor Comment 7) \* L198: What are the displacements between overlapping bins ? (i.e., spacings of final estimates in the map)**

This has now been clarified by adding:

"... each bin slid 5° in longitude and 2.5° in latitude to yield a final 5°  $\times$  2.5° longitude  $\times$  latitude grid."

Further, we have also given the grid spacings for the other bin sizes used.

**(Minor Comment 8) L201: 'and' must be removed.**

As recommended by Reviewer #1, this text has been reworded for clarification:

"... values are multi-year means of medians, and not multi-year means of arithmetic mean values."

**(Minor Comment 9) L240: 'single' values: What does this mean ?**

The expression "single values" is misleading. Sorry for that. We have reworded this sentence:

"... the PDFs should be created by normalizing all values by, for example, ..."

(Minor Comment 10) \* L240: 'normalized by the global distribution of median values': I would suggest clarifying more, with (for instance) 'spatially and temporally varying medians'. Moreover, given the context, it seems that this median field varies even within a bin of single estimation (so that the local gradient of median is taken into account). However it is not clear how this inner-bin field is made. Later in L456, it is first mentioned that an interpolation has been used. I suggest introducing this method around here (or earlier).

This point is related to Reviewer # 1, Main Comments 2 and 3, and Minor Comment (10), as well as Reviewer # 3, Minor Comment 2.

As recommended, for clarification we have used the expression "spatially and temporally varying medians". Further, in the revised manuscript, after former I.240, we have added some more reasoning why we apply normalization, and the normalization procedure is described in detail. The importance of normalization is now also demonstrated in an Appendix where global distributions of Gini coefficients are shown that are calculated for SABER gravity wave absolute momentum fluxes without normalization. In these distributions spurious enhancements of intermittency can be seen that are introduced by strong horizontal gradients of the global distribution of momentum fluxes.

**(Minor Comment 11) L245: 'sensitivities': instrumental sensitivities ?**

The expression "sensitivities" here is meant to include both instrumental sensitivities and the effect that models may resolve only a certain portion of the gravity wave spectrum. For clarification, we have reworded this statement as follows:

"Normalization of the PDFs makes also sense if only the shapes of the PDFs of different data sets shall be compared, but magnitudes are different. In the case of observations, this could happen if instruments have different observational filters for observing gravity waves. In the case of model data, differences in magnitude could arise from different model resolutions, or from different model setups. Using normalization, it is even possible to compare completely different physical parameters that have different physical units."

(Minor Comment 12) L247-250: (just a question) In other words, momentum flux distributions show an interannual difference only in their overall magnitudes but not in the shapes of distributions. What would this mean ? Is this a statistical nature of gravity waves ?

Indeed, this could hint at the statistical nature of the gravity wave distribution. The shape of the distribution seems to not change much from year to year.

No changes required.

(Minor Comment 13) L253: 'contribute to ... for the unnormalized values': It would be more clear to say 'cause an overestimation of ... when the unnormalized values are used.'

Modified as requested.

**(Minor Comment 14) L256: 'very close to unity': Why not exactly the unity ?**

Since we are using "local" medians for normalization, and not the median of all data points in a considered region, it is not expected that the overall median of the normalized PDF in that region is exactly unity. For example, one effect of our method to obtain time-varying global distributions of medians, which uses overlapping lon/lat bins, is that at the boundaries of these larger regions there will always be lon/lat bins that are not completely contained in the larger PDF-region. This means that also information from data points outside the larger PDF-region is used for normalizing the data points inside the larger region, which can lead to minor deviations of the median of the normalized PDF inside the larger region from unity. We consider this a very minor effect that is almost negligible, and therefore we just add a short statement in the text as follows:

"Since we are using local medians for normalization, and not the overall median of all data points used for a PDF, it is not expected that the overall median of a normalized PDF is exactly unity."

(Minor Comment 15) L256: 'the reciprocal of': This might be removable in the context.

Removed as recommended.

(Minor Comment 16) L270: Here I had wondered about what the meaningful range in the left-end of the distribution would be, considering some measurement noises. Then later this information was found in Section 4.2.4 (it is so nice to have this). I would suggest referring to that section here and, in case it is possible, providing the meaningful range in values (from about -2 in most cases ?) briefly.

As requested, we will add the following statement:

"As will be shown later in Sect. 4.2.4, measurement noise only partly affects the PDFs at low values of normalized momentum fluxes. The meaningful range of the PDFs for normalized momentum fluxes starts from about -2 in most cases."

(Minor Comment 17) L272: 'the SABER PDFs in Figs. 6c and 6d': These panels are for HIRDLS.

Thank you very much for finding this mistake!

Correct is: "Figs. 6e and 6f"! This will be corrected in the revised manuscript!

**(Minor Comment 18) L276: '90th and': This should be removed, as the 90th percentile shows a slight increase in SABER.**

Corrected as requested.

(Minor Comment 19) L287: 'regions. The locations of ... are illustrated' can be shortened by 'regions illustrated'.

done

(Minor Comment 20) \* L299-300: '... wider': How can they be compared with different units ?

Thank you very much for pointing this out! We have rephrased this sentence as follows:

The intermittency of unnormalized momentum fluxes is only generally somewhat stronger as can be seen from the percentages of momentum fluxes beyond the 90th and 99th percentiles.

(Minor Comment 21) \* Table 3: '0.61 mPa' (1st row, 1st column): This value differs from Fig. 7a (0.51 mPa).

Thank you very much for finding this typo!

The value in Table 3 has been corrected to 0.51 mPa.

(Minor Comment 22) L302-305: Here the heights of the distribution tails relative to the lognormal functions (fitted for each distribution) are used for the comparison of the intermittency in the tropics and that in winter high latitudes. However, it should also be considered that the fitted lognormal function in the tropics has a width ( $2 \text{ at } 10^{-5}$ ) that is not larger than that in the high latitudes ( $2.4 \text{ at } 10^{-5}$ ). If this were not the case, the statement L302-305 would not simply hold.

Thank you very much for pointing this out. We have included this fact in the revised manuscript:

"Together with the somewhat reduced width of the fitted lognormal distribution in the tropics, this means..."

**(Minor Comment 23) L306: '90th percentiles': of unnormalized fluxes ?**

Sorry, our statement was incomplete and therefore wrong. Correct is:

"Also the percentages of momentum fluxes beyond the 90th percentiles are in good agreement ..."

**(Minor Comment 24) L325: 'balloon' is missing.**

added

(Minor Comment 25) L366: 'jet-related gravity source processes act more continuously' (in the northern hemisphere): Would this be contradictory to the fact that the mean momentum flux (unnormalized) is only less than half that in Southern Ocean in austral winter (Table 3) ?

Although gravity amplitudes are smaller on average in the Northern Hemisphere, gravity wave sources could still act more continuously. Main reason are the background winds that are weaker in the northern hemisphere polar vortex, which makes gravity wave propagation conditions less favorable and limits the maximum amplitudes that gravity waves can attain.

We have added this explanation as follows in the revised manuscript:

"Given the strong activity of jet-related gravity waves sources, it may appear counter-intuitive that gravity wave amplitudes in the Northern Hemisphere during boreal winter are much lower than in the Southern Hemisphere during austral winter. However, during the respective winter season, background winds are usually much weaker in the Northern Hemisphere than in the Southern Hemisphere. This makes gravity wave propagation conditions less favorable in the Northern Hemisphere during boreal winter and limits the maximum amplitudes that gravity waves can attain. Another effect that can lead to less pronounced hotspots of mountain wave activity in the Northern Hemisphere is that...

**(Minor Comment 26) L390: 'the regional difference in the potential energy PDFs ... to the corresponding difference in ...' ?**

In order to clarify our statement, we have reworded it as follows:

"the shapes of the potential energy PDFs in the different regions are very similar to the shapes of the corresponding momentum flux PDFs."

**(Minor Comment 27) L408: 'HIRDLS and SABER' can be deleted (repetitive).**

deleted

**(Minor Comment 28) L457-458: Has the reason for this been mentioned before ?**

No, so far we did not give any explanation for this similarity of the global distributions of Gini coefficients.

One possible reason for the similarity between median and mean normalization could be that the median differs from the mean by a factor that is approximately the same within any of the given lon/lat bins used for calculating the global maps, while this factor may vary from bin to bin. In this case, the Gini coefficient determined for this bin would be almost unaffected. We have added this explanation in the revised manuscript as follows:

"One possible reason for this similarity could be that for every lon/lat bin used for calculating the global distributions of  $I_g$ , the median differs from the mean by a factor that does not vary much within this bin. If this is the case, the Gini coefficient for this bin would be almost the same for both kinds of normalization. Still, the factor between mean and median could vary from bin to bin without having much effect on the global distribution of  $I_q$ ."

(Minor Comment 29) L567: 'introduce additional': I would suggest changing this to 'alter' (or 'increase/decrease'), as the temporal changes of the atmosphere can also reduce the intermittency depending on the situation (Kim et al., 2021, JAS).

Thank you very much for this additional information! The text has been changed to:

"...alter intermittency (either increase or decrease, e.g., Kim et al., 2021)"

(Minor Comment 30) L580: '... with altitude, mainly in the mesosphere.' ? (In the stratosphere, the increase seems to be very small: 0.01–0.02?)

this information has been added in the revised manuscript

(Minor Comment 31) L614: 'We will ...': Section 6.1 also focussed on SABER. Please place this sentence to the earlier part.

As suggested, this statement has been moved to the introductory part of Sect. 6.

(Minor Comment 32) L620: 'seen'

corrected

(Minor Comment 33) L642: 'gravity wave' (the first-appearing one) can be deleted.

first appearance deleted

(Minor Comment 34) L734: 'flat' at around what value (could you please include this)?

In the revised manuscript we have added "and displays only weak variations around a value of 0.5."

---

## Author Comment (AC4)

We would like to thank Referee #3 for appreciating our efforts and for the very helpful comments and suggestions that will definitively improve the manuscript!

Please find below our point-by-point reply to the reviewer concerns. Comments by Reviewer #3 are given in red, our reply is given in black, and changes in the manuscript are indicated in blue.

**Reply to the Minor Comments by Reviewer # 3:**

**(Minor Comment 1:) Convective waves: The paper mentions quite thoroughly the limitations of the satellite dataset, notably in terms of spectral characteristics of the waves that can be observed. I would appreciate though a further discussion regarding gravity waves generated by convective systems in the tropics. Like convection, the activity of those waves likely presents a strong diurnal cycle (cf. e.g., Corcos et al., 2021, reference already cited). I wonder how the HIRDLS/SABER observation characteristics (e.g., local time of passage) might alter the retrieval of the intermittency. My impression is that undersampling the diurnal cycle would probably lower the observed intermittency... I also wonder whether this might be one possible reason for the higher intermittencies reported in SABER observations around lines 515 onward.**

Of course, the local time coverage could have some effect on the intermittency and could explain the minor differences between HIRDLS and SABER intermittency in the tropics. While HIRDLS observes at two fixed local times, the SABER local time varies during one month. However, based on satellite data, local time variations of gravity waves in the tropics have not been investigated so far. One of the reasons is that the situation is even more complicated by the differences in the viewing geometry between both satellite instruments on the one hand, and the different viewing direction during the ascending and descending orbit legs on the other hand. Addressing these effects is very difficult and beyond our current study.
This means that these effects should be considered as remaining uncertainty and potential systematic error.
In the revised manuscript we will mention that such effects can lead to systematic errors and contribute to the remaining uncertainty of our results. In the revised manuscript we have added another paragraph after former l. 519.

"One effect that may play a role are the different line of sight orientations of the HIRDLS and SABER instruments (see, for example, Trinh et al., 2015). These differences will lead to different sensitivities of observing gravity waves of a given orientation. In addition, the lines of sight are different for ascending and descending satellite orbits, which may lead to systematic differences between gravity wave amplitudes and momentum fluxes detected during ascending and descending orbits, respectively. These differences are probably one of the reasons why the diurnal cycle of convectively generated gravity waves in the tropics has not been investigated so far using satellite data. Of course, this diurnal cycle will also contribute to the level of intermittency in the monthly values shown in our study. However, the above mentioned effects are difficult to quantify and should be considered as one of the remaining uncertainties."

**(Minor Comment 2) PDF Normalization process: I agree with the justification of PDF normalization... but I am still unsure how the normalization is actually applied. Actually, this is explained in only one sentence (l 239-240), which is repeated in line 247-248. I would appreciate some further details. In particular, my current understanding is that, in every gegraphical box used to obtain gravity-wave momentum flux PDFs (and for every calendar month), one normalization factor is computed: this would explain**

**that the authors refers to the "global distribution median values", which are used as normalization factors. What I feel confusing is the application of this process that is made in Figure 6 (namely Figure 6a =¿ 6b). A natural choice (at least for me!) would have to use a single normalization factor per year for the whole 65S-50S region. But this seems at odds with several of your results/remarks, so I have inferred that this is not what was applied.**

This point is related to Reviewer # 1, Main Comments 2 and 3, and Additional Comment (8), as well as Reviewer # 2, Minor Comment (10).

As recommended, we have added a detailed description of the normalization procedure in the revised manuscript after former l.240.

Applying a single normalization factor for a given year and a region as large as the whole 65S-50S region would not be sufficient because horizontal gradients of the global distribution of gravity wave potential energies or momentum fluxes can introduce spurious intermittency. The problem is that by forming a PDF (or quantifying intermittency in another way) one assumes that all data points considered follow the same distribution with the same mean and the same standard deviation. This, however, is clearly not the case if there are spatial gradients caused by variations of the overall global distribution within an area considered. These variations are compensated for by normalizing values by the temporally and spatially varying global distribution of medians.

This reasoning is also given in the revised manuscript after former l.240.
Further, in the revised manuscript, the importance of spatially and temporally varying normalization is now demonstrated in an Appendix where global distributions of Gini coefficients are shown that are calculated for SABER gravity wave absolute momentum fluxes without normalization. In these distributions spurious enhancements of intermittency can be seen that are introduced by strong horizontal gradients of the global distribution of momentum fluxes, particularly at mid to high southern latitudes during austral winter. These enhancements are evident, even though the lon/lat bins used for gridding these global distributions are much smaller than the regions used for creating the PDFs (e.g., the whole 65S-50S region).

**Reply to the Technical Remarks by Reviewer # 3:**

**(Remark 1)** l 122: remove "for about 60 days"

done

**(Remark 2)** l 153: flying northward rather than "southward", right?

Thank you very much for finding this mistake!

corrected!

**(Remark 3)** l 184: discarded rather than "neglected"?

corrected!

**(Remark 4)** Figures 17 and 18: I would recommend putting the highest altitudes at the top of the figures and the lowest altitudes at the bottom.

As recommended, Figs. 17 and 18 were rearranged.